# The identification of extensive samples of motor units in human muscles reveals diverse effects of neuromodulatory inputs on the rate coding

Simon Avrillon[1,2]*, François Hug[3,4], Roger M Enoka[5], Arnault HD Caillet[1], Dario Farina[1]*

[1]Department of Bioengineering, Faculty of Engineering, Imperial College London, London, United Kingdom; [2]Nantes Université, Laboratory 'Movement, Interactions, Performance', Nantes, France; [3]Université Côte d'Azur, LAMHESS, Nice, France; [4]The University of Queensland, School of Biomedical Sciences, Brisbane, Australia; [5]Department of Integrative Physiology, University of Colorado Boulder, Boulder, United States

## eLife Assessment

Leveraging state-of-the-art experimental and analytical approaches, this **important** study characterizes the recruitment and activation of large populations of human motor units during slow isometric contractions in two lower limb muscles. Evidence for the main claims is **solid** and advances our understanding of how humans generate and control voluntary force.

**\*For correspondence:**
s.avrillon@imperial.ac.uk (SA);
d.farina@imperial.ac.uk (DF)

**Competing interest:** The authors declare that no competing interests exist.

**Abstract** Movements are performed by motoneurons transforming synaptic inputs into an activation signal that controls muscle force. The control signal emerges from interactions between ionotropic and neuromodulatory inputs to motoneurons. Critically, these interactions vary across motoneuron pools and differ between muscles. To provide the most comprehensive framework to date of motor unit activity during isometric contractions, we identified the firing activity of extensive samples of motor units in the tibialis anterior (129 ± 44 per participant; n=8) and the vastus lateralis (130 ± 63 per participant; n=8) muscles during isometric contractions of up to 80% of maximal force. From this unique dataset, the rate coding of each motor unit was characterised as the relation between its instantaneous firing rate and the applied force, with the assumption that the linear increase in isometric force reflects a proportional increase in the net synaptic excitatory inputs received by the motoneuron. This relation was characterised with a natural logarithm function that comprised two stages. The initial stage was marked by a steep acceleration of firing rate, which was greater for low- than medium- and high-threshold motor units. The second stage comprised a linear increase in firing rate, which was greater for high- than medium- and low-threshold motor units. Changes in firing rate were largely non-linear during the ramp-up and ramp-down phases of the task, but with significant prolonged firing activity only evident for medium-threshold motor units. Contrary to what is usually assumed, our results demonstrate that the firing rate of each motor unit can follow a large variety of trends with force across the pool. From a neural control perspective, these findings indicate how motor unit pools use gain control to transform inputs with limited bandwidths into an intended muscle force.

## Introduction

Human muscles are versatile effectors producing forces that span several orders of magnitude. They allow humans to perform a broad range of motor tasks with the same limbs, such as a surgeon closing an incision or a climber who grasps supports on a cliff. This versatility relies on a unique structure that converts neural inputs into muscle force; i.e., the motor unit, which comprises an alpha motoneuron and the muscle fibres it innervates (*Sherrington, 1925*; *Heckman and Enoka, 2012*). The nervous system controls muscle force by modulating the number of active motor units (recruitment) and their firing rates (rate coding; *Enoka and Duchateau, 2017*). This control strategy involves projecting a substantial level of common excitatory synaptic inputs to motoneurons (*Farina et al., 2014b*; *Farina and Negro, 2015*; *Negro et al., 2016b*) that recruits them in a fixed order following the size principle (*Henneman, 1957*).

Although we have a clear understanding of the general strategies used to control muscle force, we still lack a full picture of the detailed organisation of the firing activities of motor units. One of the key challenges in the field is the ability to identify the concurrent firing activity of many motor units spanning the range of recruitment thresholds observed in human muscles (*Farina and Holobar, 2016*). Indeed, much of our knowledge on the modulation of motor unit activity still derives from animal preparations (*Hounsgaard et al., 1988*; *Bennett et al., 1998b*; *Lee and Heckman, 2000*) or from serial recordings of a few concurrent motor units in humans (*Desmedt and Godaux, 1977a*; *Bigland-Ritchie et al., 1983*; *Oya et al., 2009*; *Kirk et al., 2016*), mainly identified over narrow force ranges (*Willem Monster and Chan, 1980*; *Fuglevand et al., 2015*; *Johnson et al., 2017*; *Revill and Fuglevand, 2017*).

Animal studies have shown two stages in the rate coding of individual motor units to increase muscle force (amplification and rate limiting) in response to linear increases in the net excitatory synaptic inputs (*Bennett et al., 1998a*; *Lee and Heckman, 2000*; *Powers and Binder, 2001*; *Binder et al., 2020*). These stages represent the different responses of motoneurons to ionotropic inputs due to modulation of their intrinsic properties by metabotropic inputs acting on the somato-dendritic surfaces of motoneurons through several ion channels (*Powers and Binder, 2001*; *Heckman and Enoka, 2012*; *Binder et al., 2020*). Similar characteristics in firing activity have been observed in humans (*Fuglevand et al., 2015*; *Revill and Fuglevand, 2017*; *Beauchamp et al., 2023*), although these observations have been limited to small samples of motor units (typically <40) with firing activities experimentally decoded over a narrow range of submaximal forces (typically <30% maximal voluntary contraction [MVC] force; *Fuglevand et al., 2015*; *Revill and Fuglevand, 2017*; *Beauchamp et al., 2023*). It is difficult to infer from these studies a control scheme that is generalisable to an entire motor unit pool and to all contraction levels (*De Luca, 1985*).

The combination of arrays of electromyographic (EMG) electrodes with modern source-separation algorithms set the path for the identification of populations of active motor units - and their motoneurons - in humans (*Farina and Holobar, 2016*; *Negro et al., 2016a*). Recent studies have identified the concurrent firing activity of up to 60 motor units per contraction and per muscle by increasing the density of electrodes (*Muceli et al., 2022*; *Caillet et al., 2023a*). Here, we further extended this approach by tracking motor units across contractions at target forces that ranged from 10% to 80% MVC, based on the unique spatial distribution of motor unit action potentials across the array of surface electrodes (*Farina et al., 2008*; *Martinez-Valdes et al., 2017*). We were able to identify up to ~200 unique active motor units per muscle and per participant in two human muscles in vivo, yielding extensive samples of motor units that are representative of the motoneuron pools (*Caillet et al., 2023b*). With this approach, we described the non-linear transformation of the net excitatory synaptic inputs into firing activity for individual motoneurons (*Heckman and Enoka, 2012*; *Binder et al., 2020*). We specifically focused on the non-linearities between the changes in firing rate and force: amplification, saturation, and hysteresis (*Hounsgaard et al., 1988*; *Powers and Heckman, 2017*; *Binder et al., 2020*). The results provided a framework picture of the rate coding of motor units during large and slow increases and decreases in an applied isometric force.

Having access to the rate coding of motoneuron pools allowed us to understand how ionotropic and neuromodulatory inputs combine to modulate force, confirming some details of hypothetical force control schemes in humans. For example, experimental (*Wei et al., 2014*; *Naufel et al., 2019*) and computational (*Powers and Heckman, 2017*) studies have proposed a gain control mechanism driven by neuromodulatory inputs to motoneurons. One common conclusion of these studies is that

pools of motor units involved in fine motor tasks have a low gain, whereas additional motor units activated during more forceful tasks exhibit a progressively increase in gain (*Wei et al., 2014*; *Powers and Heckman, 2017*; *Naufel et al., 2019*; *Binder et al., 2020*). However, this conclusion is based on results from motor units in animals (*Powers and Binder, 2001*) and humans (*Wei et al., 2014*; *Goodlich et al., 2022*; *Henderson et al., 2022*) while experimentally modulating inputs, or by fitting a nonlinear model predicting muscle activation from excitatory synaptic inputs at multiple contraction levels (*Naufel et al., 2019*). In contrast, we addressed this question by identifying the concurrent activity of hundreds of motor units spanning most of the range of recruitment thresholds in vivo in humans.

## Results
### Identification and tracking of individual motor units

16 participants performed either isometric dorsiflexion (n=8) or knee extension tasks (n=8) while we recorded the EMG activity of the tibialis anterior (TA - dorsiflexion) or the vastus lateralis (VL - knee extension) with four arrays of 64 surface electrodes (256 electrodes per muscle). The motoneuron pools of these two muscles of the lower limb receive a large part of common input (*Laine et al., 2015*; *Negro et al., 2016b*), constraining the recruitment of their motor units in a fixed order across tasks. They are therefore good candidates for an accurate description of rate coding. Moreover, we wanted to determine whether differences in rate coding observed between proximal and distal muscles in the upper limb (*De Luca et al., 1982*) were also present in the lower limb.

The experimental tasks comprised isometric contractions with a ramp-up, a plateau, and a ramp-down. The ramp-up and ramp-down phases were performed slowly compared with contractile speeds observed during activities of daily living. They were performed at a constant rate of 5% MVC·s$^{-1}$ and the force plateau was maintained for either 10 s (70–80% MVC), 15 s (50–60% MVC), or 20 s (10–40% MVC) (*Figure 1A*). The target during the plateau ranged from 10% to 80% MVC in increments of 10% MVC (randomised order).

A source-separation algorithm (*Farina and Holobar, 2016*; *Negro et al., 2016a*) was applied to the EMG signals to extract motor unit pulse trains, from which discharge times were automatically identified. All identified motor unit firings were visually inspected and manually edited when necessary (*Avrillon et al., 2024*). Because the signal was stationary during the plateau, it was possible to estimate reliable separation vectors for large samples of motor units with source-separation algorithms (*Figure 1A*; *Farina and Holobar, 2016*; *Negro et al., 2016a*). The average number of motor units identified in each contraction per participant was 42 ± 24 (25th–75th percentile: 24–53, up to 95; *Table 1*) motor units from the TA and 33 ± 15 (25th–75th percentile: 23–47, up to 71; *Table 1*) motor units from the VL. The datasets from all target forces were merged into a sample of unique motor units per muscle and participant that spanned most of the operating range of recruitment thresholds observed in humans (1st–99th percentile: 0.9–73.4% MVC; see below).

The proportion of the EMG signal represented by the identified motor units was estimated by reconstructing a synthetic EMG signal from the firing activity. To do so, the discharge times were used as triggers to segment the differentiated EMG signals over a window of 25 ms that yielded averaged action potential waveforms for each motor unit (*Figure 1A*). The action potentials were then convolved with the discharge times to obtain trains of action potentials, and all the trains of the identified motor units were summed to reconstruct the synthetic EMG signal. The ratio between the powers of synthetic and experimental EMG signals was calculated (*Figure 1B*): it was 69.3 ± 17.3% (25th–75th percentile: 59.3–83.6%, up to 94.2%) for TA and 55.2 ± 19.5% (25th–75th percentile: 50.0–71.9%, up to 86.5%) for VL (*Figure 1C*). These values indicate that most of the recorded surface EMG signals were successfully decomposed into motor unit activity.

Motor units were tracked between contractions using their unique spatial distribution of action potentials (*Figure 1D*; *Martinez-Valdes et al., 2017*). This method was validated by using both simulated EMG signals and two-source validation with simultaneous recordings of firing activity from intramuscular and surface EMG signals (*Figure 1—figure supplement 1*). On average, we tracked 67.1 ± 10.0% (25th–75th percentile: 53.9–80.1%) of the motor units between consecutive target forces (10% increments, e.g. between 10% and 20% MVC) for TA and 57.2 ± 5.1% (25th–75th percentile: 46.6–68.3%) of the motor units for VL (*Figure 1—figure supplement 2*). There are two explanations for the inability to track all motor units across consecutive target forces: (i) some motor units are recruited

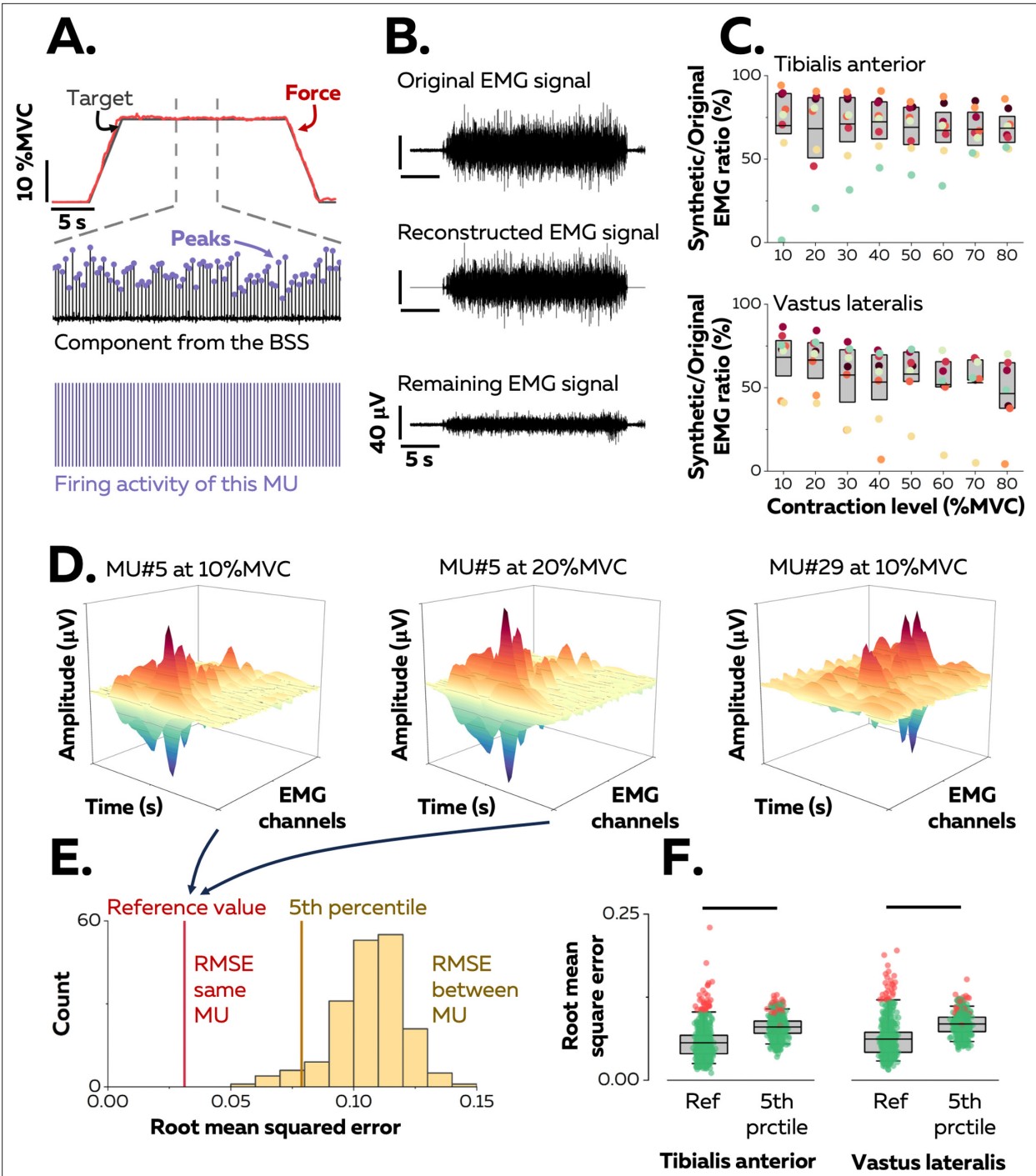

**Figure 1.** Identification of motor units in two human muscles. (**A**) We used a blind-source-separation (BSS) algorithm to decompose the overlapping activity of motor units (MU) into spike trains during a force-matching trapezoidal task (red trace). (**B**) We reconstructed synthetic electromyographic (EMG) signals by summing the trains of action potentials from all the identified motor units and interpreting the remaining EMG signal as the part of the signal not explained by the decomposition. (**C**) We calculated the ratio between the powers of synthetic and original EMG signals to estimate the proportion of the signal variance explained by the decomposition. Each data point indicates the average value for one participant. (**D**) We estimated the uniqueness of each identified motor unit within the pool by calculating the root-mean-square error (RMSE) between the distributions of action potentials of the same motor unit across contractions (two panels on the left, reference value in **E**) and between motor units (left vs. right panels, distribution of RMSE between motor units in yellow in **E**). (**F**) Each motor unit was unique within the pool when the RMSE between its distributions of action potentials across target forces (reference value) was less than the 5th percentile of the distribution of RMSE with the rest of the motor units. Motor units considered as outliers in F (red data points) were removed from the analysis due to potential errors in tracking between contractions. Each data point is a motor

*Figure 1 continued on next page*

*Figure 1 continued*

unit, the box represents 25th–75th percentiles of the distribution of data, and the black line shows the median. The horizontal thick line denotes a statistical difference between reference values and 5th percentiles for each muscle.

The online version of this article includes the following figure supplement(s) for figure 1:

**Figure supplement 1.** Validation of electromyographic decomposition and motor unit tracking.

**Figure supplement 2.** Recruitment thresholds of motor units tracked across contractions.

at higher targets only; (ii) it is challenging to track small motor units beyond a few target forces due to a lower signal-to-noise ratio when larger motor units are recruited, or signal cancellation (*Keenan et al., 2005*; *Farina et al., 2014a*). In total, we identified 129 ± 44 motor units (25th–75th percentile: 100–164, up to 194) per participant in TA and 130 ± 63 motor units (25th–75th percentile: 103–173, up to 199) per participant in VL.

The accuracy of the tracking method was further tested by confirming the uniqueness of each motor unit within the identified sample, assuming that each motor unit had a unique representation across the array of surface electrodes (*Farina et al., 2008*). This was accomplished by calculating the root-mean-square error (RMSE) between their action potentials across the electrodes relative to those of the rest of the motor units (*Figure 1E*). The RMSE between the action potentials of the same motor unit tracked across contractions was calculated as a reference and compared with the 5th percentile of the distribution of RMSE between motor units (*Figure 1F*). The reference value was typically less than the 5th percentile of the RMSE calculated with the action potentials of the other motor units (TA: 5.6 ± 2.4 vs. 8.0 ± 1.5%; p<0.001; VL: 6.2 ± 2.8 vs. 8.5 ± 1.6%; p<0.001). Inspection of the data for each motor unit revealed that 92.1% (TA) and 87.0% (VL) of the motor units had a lower RMSE between target forces compared with the other motor units (<5th percentile). Of note, motor units with the highest reference values (>95th percentile) were considered as outliers due to tracking errors and were excluded from the subsequent analyses (*Figure 1F*). We excluded 34 motor units from TA and 28 from VL.

## Non-linear rate coding during the ramp-up phase - amplification and rate limiting

The input-output function for each motoneuron was characterised as the relation between its instantaneous firing rate and the applied force during the ramp-up phase of the contractions (*Figure 2A*). The linear increase in force was assumed to reflect a proportional increase in the net synaptic excitatory inputs received by the motoneurons, as proposed previously (*Fuglevand et al., 1993*; *Revill and Fuglevand, 2017*). Thus, any deviation from a linear increase in firing rate with respect to a linear increase in force presumably reflects the influence of neuromodulatory inputs on motoneuron gain (*Johnson et al., 2017*; *Revill and Fuglevand, 2017*; *Beauchamp et al., 2023*), or a saturation of the motoneuron firing rate (*Fuglevand et al., 2015*; *Revill and Fuglevand, 2017*).

The association between the firing rate and the force was characterised by comparing three curve fits: (i) linear - an increase in firing rate proportional to the increase in the net synaptic excitatory input, (ii) rising exponential - an initial acceleration of firing rates followed by full saturation (*De Luca and Contessa, 2012*; *Fuglevand et al., 2015*; *Revill and Fuglevand, 2017*), and (iii) natural logarithm - an initial acceleration of firing rate followed by a slower constant increase in firing rate that reflects a rate limiting effect (*Figure 2A*; *Bennett et al., 1998a*; *Lee and Heckman, 2000*; *Powers et al., 2012*). Of note, this analysis was performed on each unique motor unit with the data pooled across all

**Table 1.** Mean ± standard deviation (range) for the number of motor units across the eight target forces and two muscles.

|      | 10%              | 20%              | 30%              | 40%              | 50%              | 60%              | 70%              | 80%              |
|------|------------------|------------------|------------------|------------------|------------------|------------------|------------------|------------------|
| TA   | 38 ± 25 (1–73)   | 45 ± 24 (21–83)  | 50 ± 27 (24–95)  | 49 ± 26 (24–93)  | 45 ± 27 (15–93)  | 40 ± 25 (14–80)  | 37 ± 26 (14–84)  | 34 ± 19 (9–60)   |
| VL   | 44 ± 17 (17–63)  | 46 ± 20 (19–71)  | 42 ± 20 (10–67)  | 34 ± 18 (3–56)   | 32 ± 15 (5–53)   | 26 ± 13 (3–38)   | 26 ± 14 (1–38)   | 20 ± 13 (1–37)   |

TA, tibialis anterior; VL, vastus lateralis.

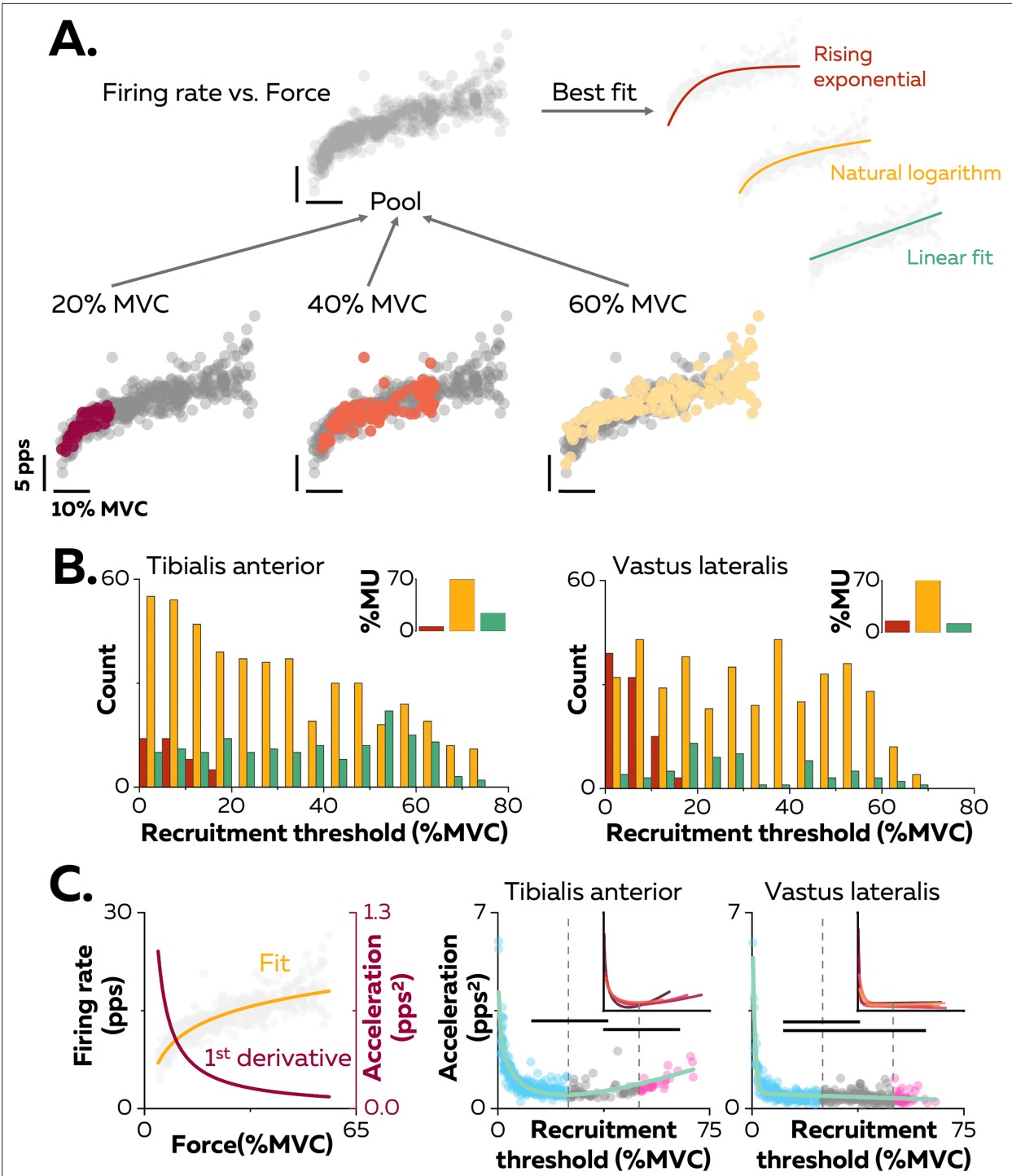

**Figure 2.** Non-linear rate coding of motor units. (**A**) The relation between firing rate (pulses per second, pps) and the applied force during the ramp-up phase of the contraction was determined by concatenating the instantaneous firing rates for each motor unit (grey data points) recorded over all the contractions where it was identified, as shown here for one motor unit (coloured data points for contractions at 20%, 40%, and 60% MVC). The derived relations were then fitted with three different functions: linear (green), rising exponential (dark red), and natural logarithm (yellow), to characterise the input-output function of each motoneuron. (**B**) The motor units were grouped according to their best fit. The graphs show the distribution of these groups as a function of recruitment thresholds (RT) for each muscle. The inset panels depict the percentage of motor units (MU) in each group. (**C**) The initial acceleration of firing rate was derived from force-firing rate relation fitted with the natural logarithm (f(force)=a*ln(force)+b; yellow trace) and its first derivative (f(force)=a/force; dark red trace). The right panels show the distribution of initial acceleration values relative to recruitment threshold (RT) for all participants (n=328 motor units for tibialis anterior [TA] and n=393 motor units for vastus lateralis [VL]). Each data point indicates a motor

*Figure 2 continued on next page*

*Figure 2 continued*

unit. The horizontal thick lines denote a statistical difference between the motor units groups (low-threshold=blue; medium-threshold=grey; high-threshold=pink). The green line depicts the non-linear fits of these relations for the TA and the VL. Similar fits were observed for all the participants (inset panels).

The online version of this article includes the following figure supplement(s) for figure 2:

**Figure supplement 1.** Relations between force and firing rates fitted with a natural logarithm.

contractions (*Figure 2A*). The result was that 69.5% of the motor units in TA and 72.3% of those in VL had an input-output function better fitted by a natural logarithm (*Figure 2B*, inset panels).

The input-output functions of the motor units were then compared based on the distribution of recruitment thresholds (1st–99th percentile: 0.9–73.4% MVC). Motor units were clustered into three groups with equal ranges of recruitment thresholds: low- (0–25% MVC), medium- (25–50% MVC), and high- (50–75% MVC) threshold motor units (*Figure 2B*). A significant percentage of input-output functions for low-threshold motor units was better fitted with a rising exponential function (12.5% for the TA, 30.9% for the VL), highlighting a steep acceleration of firing rate followed by a full saturation. This pattern was not observed in medium- and high-threshold motor units (0% for TA and VL). In contrast, a significant percentage of input-output functions of high-threshold motoneurons was better fitted with a linear function for TA (39.6%), although the input-output functions of high-threshold motor units from VL were still better fitted with natural logarithms (87.9%) than linear functions (12.1%).

We further described the first stage of rate coding by estimating the acceleration of firing rate (*Figure 2C*; *Figure 2—figure supplement 1*). The initial acceleration of the firing rate, likely due to the amplification of synaptic inputs by persistent inward currents (*Lee and Heckman, 2000*), was calculated as the value of the first derivative of the natural logarithm function at the recruitment threshold (*Figure 2C*). These values were compared between motor units using a linear mixed effect model, with *muscle* (TA; VL) and *threshold* (low; medium; high) as fixed effects and *participant* as a random effect. There was a significant effect of *muscle* (F=30.4; p<0.001), *threshold* (F=24.7; p<0.001), and a significant interaction *muscle×threshold* (F=4.1; p=0.017). The initial acceleration of firing rate in TA was greater for the low- ($1.0 \pm 0.7$ pps$^2$; p<0.001) and the high- ($1.0 \pm 0.3$ pps$^2$; p=0.032) threshold motor units than for medium-threshold motor units ($0.6 \pm 0.3$ pps$^2$), with no difference between the low- and high-threshold motor units (p=0.999; *Figure 2C*). Similarly, the initial acceleration of firing rate was greater for low-threshold motor units ($0.6 \pm 0.6$ pps$^2$) than for medium- ($0.4 \pm 0.2$ pps$^2$; p<0.001) and high-threshold motor units ($0.4 \pm 0.2$ pps$^2$; p=0.013) in VL, with no difference between the medium- and high-threshold motor units (p=0.990; *Figure 2C*). The initial acceleration of firing rate was greater for the low- and high-threshold motor units in TA compared with VL (p<0.001 for both).

We also tested whether the distribution of initial accelerations of firing rate observed across the entire motor unit pool was generalisable to all the participants. Participants with either fewer than 20 motor units or with no motor units recruited below 5% MVC were excluded from this analysis (TA: n=4; VL: n=3). The function representing the relation between recruitment thresholds and initial acceleration were accurately fitted using the non-linear least-square fitting method (TA; adjusted R-squared: 0.77; VL; adjusted R-squared: 0.80; green lines on *Figure 2C*). We then fitted the same non-linear functions for each participant; the adjusted R-squared values increased to $0.86 \pm 0.04$ for the TA and $0.84 \pm 0.11$ for the VL (inset panels in *Figure 2C*).

The second stage of rate coding involved a linear increase in firing rate after the initial acceleration phase. This stage was analysed using the average firing rate during the successive force plateaus for each tracked motor unit (*Figure 3*). Specifically, the increase in firing rate between consecutive contraction levels (i.e. change in force equal to 10% MVC) was compared with a linear mixed effect model with *muscle* (TA; VL) and *threshold* (low; medium; high) as fixed effects and *participant* as a random effect. There were significant effects for *muscle* (F=39.7; p<0.001) and *threshold* (F=145.3; p<0.001), and a significant interaction *muscle×threshold* (F=37.7; p<0.001). The rate of increase was significantly greater for TA ($3.5 \pm 1.7$ pps; 25th–75th percentile: 2.4–4.1 pps) than for VL ($2.0 \pm 1.1$ pps; 25th–75th percentile: 1.3–2.6 pps. p<0.001). The rate of increase was significantly greater in TA for high-threshold than for low- (p<0.001) and medium- (p<0.001) threshold motor units and for medium- than for low-threshold motor units (p<0.001). In contrast, the rate of increase in VL was greater for high- (p<0.001) and medium- (p<0.001) threshold motor units than for low-threshold motor units, with no difference between high- and medium-threshold motor units in VL (p=0.993).

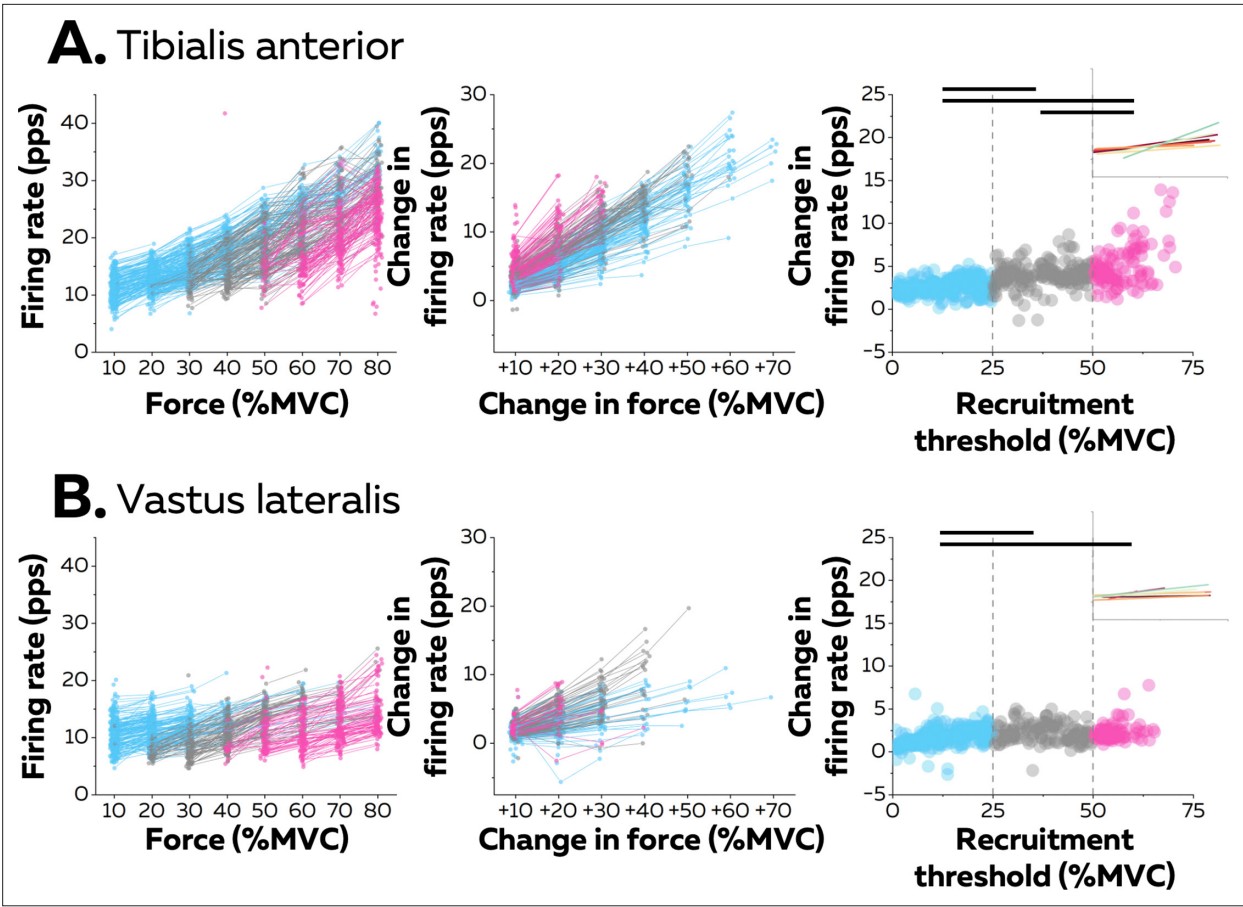

**Figure 3.** Motor unit firing rates across contraction levels. The left column shows average firing rate (pulses per second, pps) during the force plateaus for each tracked motor unit across contraction levels for all participants from tibialis anterior (TA) (n=998 motor units; **A**) and vastus lateralis (VL) (n=1016 motor units; **B**). Each data point indicates one motor unit, and each line connects the firing rates of this motor unit across contractions. The colour scale identifies the three groups of motor units based on recruitment threshold: low (blue), medium (grey), and high (pink). The middle column depicts the change in firing rates between contractions separated by 10–70% of the maximal voluntary contraction level (MVC) of force. The right column shows the relation between the rate of increase in firing rate between successive target forces (e.g. between 10% and 20% MVC) and the recruitment threshold of the motor unit. These relations were fitted with a linear function (coloured lines in the inset panels) for each participant. The horizontal thick line denotes a statistical difference between motor units grouped by recruitment thresholds.

The linear functions for each participant are shown in *Figure 3*. There was a significant positive association between the rate of increase in firing rate and the recruitment threshold for all the participants (Pearson's r: TA: 0.71 ± 0.14 pps and VL:0.57 ± 0.22 pps; all p<0.022), except in one participant for each muscle (TA: r=0.17; p=0.272; VL: r=0.12; p=0.411). Most motor units (92.0% in TA and 80.9% in VL) increased firing rates by more than one pulse per second between contractions up to the maximal force tested or until tracking was not possible.

## Differences between ramp-up and ramp-down hysteresis

Neuromodulatory inputs can prolong excitatory synaptic inputs and produce unequal recruitment and derecruitment thresholds; i.e., a hysteresis. The left columns in *Figure 4* show the relation between recruitment and derecruitment thresholds within the motor unit pools (*Figure 4*). The relation for most participants was better characterised with a non-linear than a linear function (TA: 6 out of 8 participants, adjusted R-squared>0.92; VL: 7 out of 8 participants, adjusted R-squared>0.83). The difference between recruitement and derecruitment thresholds was compared with a linear mixed effect model with *muscle* (TA; VL) and *threshold* (low; medium; high) as fixed effects and *participant* as a random effect. There was a significant effect of *threshold* (F=83.0; p<0.001) and a significant interaction *muscle×threshold* (F=30.1; p<0.001), but no effect of *muscle* (F=2.5; p<0.137). We further tested for each group of motor units whether the difference between the two thresholds was significantly

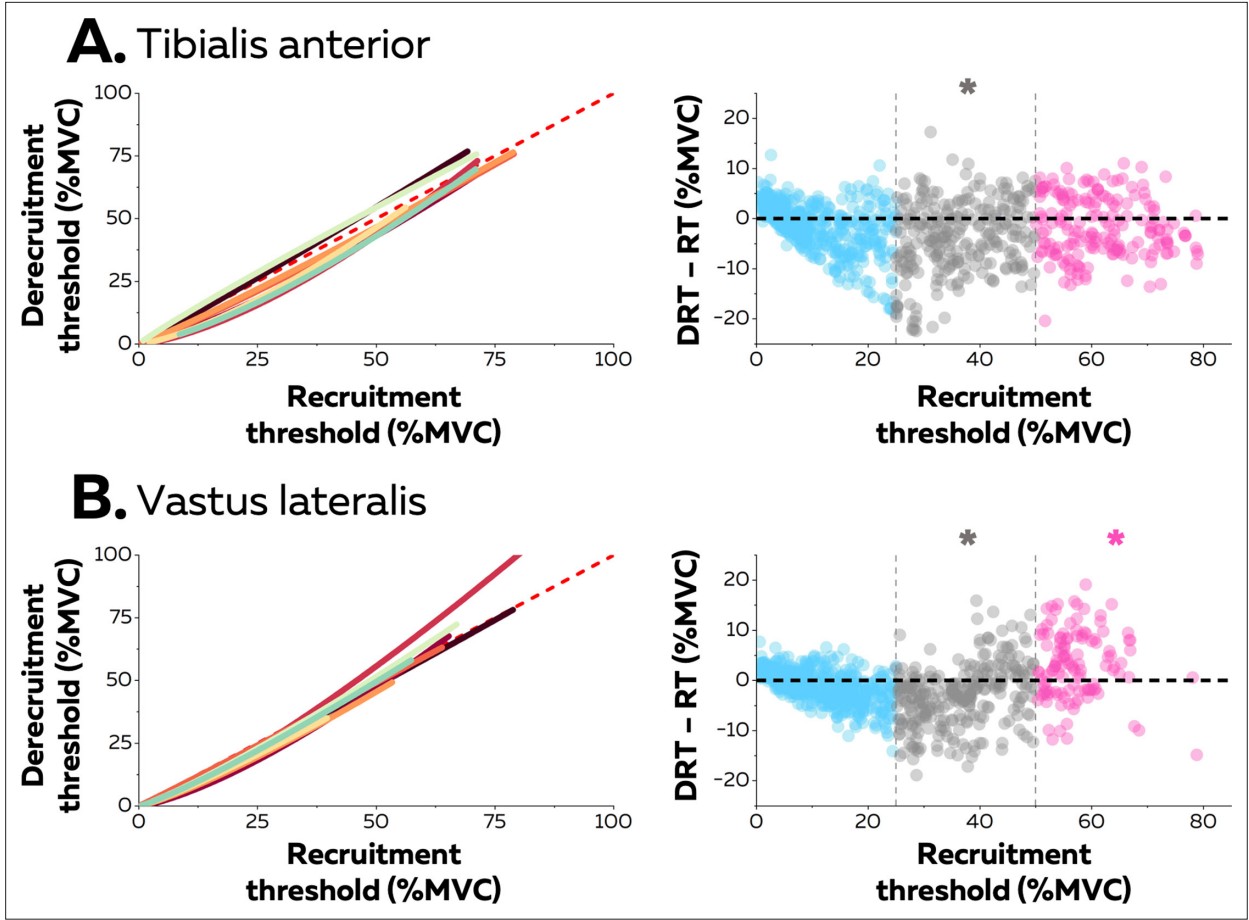

**Figure 4.** Hysteresis between recruitment and derecruitment thresholds. The left column depicts the relations between the recruitment and derecruitment thresholds of each motor unit from tibialis anterior (TA) (**A**) and vastus lateralis (VL) (**B**). These relations were fitted for each participant (coloured lines) using either non-linear or linear regressions. The values below the dashed red line (recruitment threshold = derecruitment threshold) show a positive hysteresis between recruitment and derecruitment thresholds, the values above a negative hysteresis. The right column shows the difference between the recruitment and derecruitment thresholds, with negative values showing a positive hysteresis with recruitment threshold greater than derecruitment threshold and the positive values indicating the converse (a negative hysteresis). Each data point is a motor unit. The asterisk denotes a statistical difference between the hysteresis values for motor units grouped according to recruitment threshold (low = blue; medium = grey; high = pink) and the absence of a hysteresis (dashed horizontal line).

different from 0 (absence of hysteresis). The result was that only medium-threshold motor units had a significant positive hysteresis (derecruitment threshold was lower) in both muscles (TA: –3.8 ± 7.2 pps; p=0.013; VL: –3.1 ± 6.0 pps; p=0.026), whereas low-threshold motor units from both muscles (TA: –1.5 ± 4.8; VL: –1.0 ± 3.2 pps; p>0.221) did not exhibit significant hysteresis. Although high-threshold motor units from TA followed the same trend (–1.7 ± 6.0 pps; p<0.437), high-threshold motor units from VL exhibited a negative hysteresis (VL: +3.2 ± 6.7 pps; p=0.039), which indicated that derecruitment threshold was greater than the recruitment threshold.

We also compared the type of the best fit between motor unit firing rate and force between the ramp-up and ramp-down phases of the isometric contraction. Most motor units (TA: 60.2%; VL: 76.7%) were best fit with the same relation during the two phases. Although the percentage of associations better fitted with linear functions increased for the ramp-down phase for the TA (from 25.4% to 41.9% of motor units), there was no change between ramps for the VL (from 12.8% to 19.4% of motor units). Thus, most motor units kept a non-linear relation during the ramp-down phase.

## Discussion

No previous studies have identified to date the firing activity of >100 motor units spanning most of the spectrum of recruitment thresholds from the same individual, whether in animals or humans. According to previous estimates, the number of motor units detectable with surface EMG in human would be 200 ± 61 for TA and 146 ± 29 for VL (*Duchateau and Enoka, 2022*). We have therefore identified in this study most of the active motor units present in the recorded volume for each muscle (*Figure 1*), which has allowed us to infer the rate coding of pools of motor units during slow isometric contractions. Overall, we found that motor units within a pool exhibit distinct rate coding with changes in force level (*Figures 2 and 3*), which contrasts with the long-held belief that rate coding is similar across motor units from the same pool (*Fuglevand et al., 1993*; *De Luca and Contessa, 2012*).

The fast initial acceleration stage reflects the amplification of synaptic inputs through the activation of persistent inward currents via voltage-gated sodium and calcium channels (*Bennett et al., 1998a*; *Lee and Heckman, 2000*; *Heckman and Enoka, 2012*; *Binder et al., 2020*). Although noradrenergic and serotonergic synapses that generate persistent inward currents are widespread within motor unit pools (*Maratta et al., 2015*), the influence of the neuromodulatory inputs is greater during this phase for low- than for high-threshold motor units. This is presumably due to a larger ratio between the amplitude of inward currents and the input conductance in low-threshold motoneurons (*Huh et al., 2017*). The level of recurrent and reciprocal inhibition has also probably increased with the increase in force during the ramp-up, progressively blunting the effect of persistent inward currents for late-recruited motor units (*Kuo et al., 2003*; *Hyngstrom et al., 2007*; *Revill and Fuglevand, 2017*). This may also explain the larger percentage of high-threshold motor units with a linear fit for the firing rate/force relation (*Figure 2*), as the integration of larger inhibitory inputs should linearise the firing rate/force relation (*Revill and Fuglevand, 2017*).

The second stage of rate coding involves a slower linear increase in firing rate, as characterised by the second part of a logarithmic function (*Figures 2–3*). This stage corresponds to mechanisms previously referred to as 'rate limiting' (*Heckman and Binder, 1993*; *Heckman and Enoka, 2012*; *Powers and Heckman, 2017*) or 'saturation' (*De Luca and Contessa, 2012*; *Fuglevand et al., 2015*), and was more pronounced for low- than high-threshold motor units (*Figure 3*). This difference may be explained by smaller excitatory synaptic inputs onto low- than high-threshold motoneurons (*Powers and Binder, 2001*; *Heckman and Enoka, 2012*), lower synaptic driving potential of the dendritic membrane (*Powers and Binder, 2000*; *Cushing et al., 2005*; *Fuglevand et al., 2015*), and longer and larger afterhyperpolarisation phase in low- than high-threshold motoneurons (*Bakels and Kernell, 1993*; *Gardiner, 1993*; *Deardorff et al., 2013*; *Caillet et al., 2022*).

Taken together, these results show how ionotropic and neuromodulatory inputs to motoneurons uniquely combine to generate distinct rate coding across the pool, even if a more direct manipulation of the sources of neuromodulatory and ionotropic inputs will be required to directly estimate their interactions. While the size of the motor unit determines its recruitment threshold, neuromodulatory inputs determine its gain. Thus, in a similar fashion as size imposes a fixed recruitment order (*Henneman, 1957*), neuromodulatory inputs determine a spectrum of variations in motor unit firing rates without the need to differentiate the ionotropic inputs to each motoneuron. (*Figure 2C*; *Figure 2—figure supplement 1*; *Figure 3*; *Powers and Heckman, 2017*; *Bräcklein et al., 2022*; *Chardon et al., 2024*; *Škarabot et al., 2023b*). Because high-threshold motor units exhibit a higher gain than low-threshold motor units (*Figure 3*), their firing rate could eventually reach similar or even greater values than that of low-threshold motor units during strong contractions (*Gydikov and Kosarov, 1974*; *Moritz et al., 2005*; *Oya et al., 2009*; *Škarabot et al., 2023b*). This indicates that the onion skin principle (*De Luca and Contessa, 2012*; *De Luca and Contessa, 2015*) may not hold in all muscles and for all contraction forces (*Gydikov and Kosarov, 1974*; *Moritz et al., 2005*; *Oya et al., 2009*; *Škarabot et al., 2023b*). In addition, rate coding patterns should also vary with the pattern of contractions, with fast contractions lowering the range of recruitment thresholds within motoneuron pools (*Desmedt and Godaux, 1977b*; *Desmedt and Godaux, 1979*; *van Bolhuis et al., 1997*). The variability in rate coding observed here between motor units from the same pool could lead to small deviations from the size principle sometimes observed between pairs of units during isometric contractions with various patterns of force (*Desmedt and Godaux, 1979*; *Marshall et al., 2022*) or during the derecruitment phase (*Bräcklein et al., 2022*).

It is also worth noting that the profile of rate coding mostly followed the same trend during the ramp-up and ramp-down phases of the contraction, but with prolonged firing activity only evident for medium-threshold motor units (*Figure 4*). This hysteresis is possible because of the bistability of the membrane potential mediated by inward currents from calcium channels, the two stable membrane states being the resting potential, and a depolarised state that enables self-sustained firing (*Hounsgaard et al., 1988*; *Lee and Heckman, 1998*). The absence of statistically significant hysteresis in low-threshold motoneurons may be simply explained by their early recruitment during the first percentages of force, mathematically lowering the potential amplitude of their hysteresis. Alternatively, prominent outward currents may attenuate the impact of persistent inward currents on the prolongation of their firing activity (*Powers and Heckman, 2015*). Similarly, the absence of hysteresis (TA) and even a negative hysteresis (VL) in high-threshold motoneurons can be explained by a briefer membrane bistability due to a faster decay of inward currents from calcium channels, and a narrower range of membrane depolarisation over which these inward currents are activated (*Lee and Heckman, 1998*). This result must be confirmed with a more direct proxy of the net synaptic drive, such as the firing rate of a *reference* low-threshold motor neuron used in the delta F method (*Gorassini et al., 1998*), or the cumulative spike train of low-threshold motor neurons (*Afsharipour et al., 2020*).

The increase in firing rate was also significantly greater for TA motor units than for those in VL. This difference may reflect a varying balance between excitatory/inhibitory synaptic inputs and neuromodulation due to multiple spinal circuits (*Heckman and Binder, 1993*; *Heckman et al., 2008*; *Johnson et al., 2017*; *Powers and Heckman, 2017*; *Chardon et al., 2024*; *Škarabot et al., 2023b*). Specifically, the strength of recurrent and reciprocal inhibitory inputs to motoneurons innervating VL and TA, and their proportional or inverse covariation with excitatory inputs, respectively, may explain the differences in rate limiting and maximal firing rates (*Heckman and Binder, 1993*; *Heckman et al., 2008*; *Johnson et al., 2017*; *Powers and Heckman, 2017*; *Chardon et al., 2024*; *Škarabot et al., 2023b*). Thus, the motor units from the VL may receive more recurrent inhibition than those of distal muscles, though direct evidence of these differences remains to be found in humans (*Windhorst, 1996*). Interestingly, similar differences in rate coding were previously observed between proximal and distal muscles of the upper limb (*De Luca et al., 1982*). However, other muscles that serve different functions within the human body, such as muscles from the face, have different rate coding characteristics with much higher firing rates (*Kirk et al., 2021*). Future work should investigate those muscles and other to reveal the myriads of rate coding strategies in human muscles.

Our results on rate coding characteristics of the motor unit pool provide insight on force control strategies. The accurate control of force depends on the firing activity of the population of active motor units, which effectively filters out the synaptic noise of individual motor units and primarily transforms common synaptic inputs into muscle force (*Dideriksen et al., 2012*; *Farina et al., 2014b*; *Farina and Negro, 2015*). Because the bandwidth of muscle force is <10 Hz during isometric contractions (*Enoka and Farina, 2021*), the activation of inward currents to low-threshold motoneurons at recruitment may enable them to promptly discharge at a rate (>8 pps) that transmits the effective common synaptic inputs (<8–10 Hz) without phase distortion (*Dideriksen et al., 2012*; *Farina et al., 2014b*; *Farina and Negro, 2015*). This mechanism facilitates the accuracy of force control.

Moreover, force generation can be described as the filtering of the cumulative firing activity of active motor units with the average twitch force of their muscle units (*Farina and Negro, 2015*; *Thompson et al., 2018*). From this perspective, at low forces, recruitment of new motor units has a higher impact on force modulation than rate coding since each recruitment impacts both the cumulative firing activity of the pool and the average twitch force. Rate coding only modulates the cumulative firing activity. Thus, the amplification of the firing rate of low-threshold motor units near their recruitment threshold instantly favours rate coding over the recruitment of additional motor units, which likely allows for smoother force control. Similarly, the progressive recruitment of motor units with a higher gain promotes faster changes in firing rates, which promotes force control accuracy across the full force range. On a different note, the steep increase in firing rate over the first percentages of the ramp-up may also enable the motor units to produce the required level of force despite having a more compliant muscle-tendon unit (*Mazzo et al., 2021*). Overall, our results may help to design future non-linear decoders that aim to predict muscle activation or muscle force from descending

inputs recorded in the motor cortex, with the will to generalise their performance across movements (*Naufel et al., 2019*).

## Materials and methods

### Participants

16 young individuals volunteered to participate either in the experiment on the TA (n=8; age: 27 ± 3) or on the VL (n=8; age: 27 ± 10). They had neither history of lower limb injury in the last 6 months before the experiments nor lower leg pain that would impair their ability to complete the experimental tasks. The study was reviewed and approved by Imperial College London (Study 18IC4685) and Comité de Protection des Personnes Ouest III (Study 23.00453.000166) and followed the standards of the declaration of Helsinki. Participants provided their informed written consent before starting the experimental session.

### Experimental setup

The two experimental sessions consisted of either a series of submaximal isometric ankle dorsiflexions or isometric knee extensions. EMG signals were recorded from either the TA or the VL muscles using four arrays of 64 surface electrodes for a total of 256 electrodes.

For the session of ankle dorsiflexions, participants sat on a massage table with the hips flexed at 45°, 0° being the hip neutral position, and the knees fully extended. The foot of the dominant leg (right in all participants) was fixed onto the pedal of an ankle dynamometer (OT Bioelettronica, Turin, Italy) positioned at 30° in the plantarflexion direction, 0° being the foot perpendicular to the shank. The thigh and the foot were fixed with inextensible Velcro straps. Force signals were recorded with a load cell (CCT Transducer s.a.s, Turin, Italy) connected in-series to the pedal using the same acquisition system as for the EMG recordings (EMG-Quattrocento; OT Bioelettronica, Italy).

For the session of knee extensions, participants sat on an instrumented chair with the hips flexed at 85°, 0° being the hip neutral position, and the knees flexed at 85°, 0° being the knees fully extended. The torso and the thighs were fixed to the chair with Velcro straps and the tibia were positioned against a rigid resistance connected to force sensors (Metitur, Jyvaskyla, Finland). The force signals were recorded using the same acquisition system as for the EMG recordings.

The experimental session began with a warm-up that consisted of submaximal isometric contractions from 50% to 80% of the subjective MVC. Then, participants performed two MVC with 2 min of rest in between. The maximal torque was considered as the highest torque value recorded over an average window of 250 ms. The rest of the session consisted of a series of eight submaximal isometric contractions performed at 10–80% of the MVC with 10% MVC increments. The pattern of the contractions followed a trapezoidal target displayed on a screen in real time with the force trace. Ramp-up and ramp-down phases were performed at a constant pace of 5% MVC·s⁻¹. The force plateaus were maintained for either 10 s (70–80% MVC), 15 s (50–60% MVC), or 20 s (10–40% MVC). The contractions were separated by 90 s of rest and their order was randomised.

### EMG recording

Surface EMG signals were recorded from the TA or the VL using 4 two-dimensional arrays of 64 electrodes (GR04MM1305 for the TA; GR08MM1305 for the VL, 13×5 gold-coated electrodes with one electrode absent on a corner; interelectrode distance: 4 and 8 mm, respectively; OT Bioelettronica, Italy). The grids were positioned over the muscle bellies to cover the largest surface while staying away from the boundaries of the muscle identified by manual palpation. Before placing the electrodes, the skin was shaved and cleaned with an abrasive pad and water. A biadhesive foam layer was used to hold each array of electrodes onto the skin, and conductive paste filled the cavities of the adhesive layers to make skin-electrode contact. For two-source validation of the EMG decomposition, an intramuscular linear array of 40 electrodes on a thin-film (platinum coated, interelectrode distance: 1 mm) was inserted into the TA in one participant at an approximate angle of 30°. The insertion was guided with a portable ultrasound probe (Butterfly IQ+, Butterfly Network, USA). Two bands damped with water were placed around the ankle as ground and reference electrodes. EMG signals were recorded in monopolar derivation with a sampling frequency of 2048 Hz for surface electrodes, 10,240 Hz for intramuscular electrodes, amplified (×150), band-pass filtered (10–500 Hz for surface; 100–4400 Hz for

intramuscular), and digitised using a 400 channels acquisition system with a 16-bit resolution (EMG-Quattrocento; OT Bioelettronica, Italy).

## HD-EMG decomposition

The monopolar EMG signals were band-pass filtered between 20 and 500 Hz with a second-order Butterworth filter, and channels with low signal-to-noise ratio or artefacts were discarded after visual inspection. The EMG signals were decomposed into individual motor unit pulse trains using convolutive blind source separation (*Negro et al., 2016a*). EMG signals were first extended by adding delayed versions of each channel, with an extension factor, i.e., number of delayed versions, adjusted to reach a total of 1000 extended channels. The extended signals were spatially whitened to make them uncorrelated and of equal power. Thereafter, a fixed-point algorithm was applied to identify the sources of the EMG signals, i.e., the motor unit pulse trains. In this algorithm, a contrast function g(x)=log(cosh(x)) and its derivatives were applied to the extended and whitened EMG signals to iteratively optimise a separation vector that skew the distribution of the values of the motor unit pulse trains towards 0, and thus maximise the level of sparsity of the motor unit pulse train. The high level of sparsity matches the physiological properties of motor units, with a relatively small number of discharges, or 1, per second (<50 discharge times·s$^{-1}$ during submaximal isometric contractions). The convergence was reached once the level of sparsity did not substantially vary (with a tolerance fixed at 10$^{-4}$) when compared to the previous iteration. At this stage, the motor unit pulse train contained high peaks (i.e. the discharge times of the identified motor unit) and lower values due to the activities of neighbouring motor units and noise. High peaks were separated from lower values using the MATLAB functions *findpeaks.m* and *kmeans.m* (with two classes). The peaks from the class with the highest centroid were considered as the discharge times of the identified motor unit.

After this automatic identification of the discharge times, duplicates were automatically removed. For this purpose, the pulse trains of pairs of motor units were first aligned using a cross-correlation function to account for a potential delay due to the propagation time of action potentials along the fibres. Then, two discharge times identified from these pulse trains were considered as common when they occurred within a time interval of 0.5 ms, and two or more motor units were considered as duplicates when they had at least 30% of their identified discharge times in common (*Holobar et al., 2010*; *Negro et al., 2016a*). In principle, the limited level of synchronisation between individual motor units results in a few simultaneous discharge times between pairs of motor units. A threshold of 30% is therefore highly conservative and ensure the removal of all motor units with a level of synchronisation well above physiological values. When duplicates were identified, the motor unit with the lowest coefficient of variation of the inter-spike intervals was retained for the analyses.

At the end of these automatic steps, all the motor unit pulse trains and identified discharge times were visually inspected, and manual editing was performed to correct the false identification of artefacts or the missed discharge times (*Del Vecchio et al., 2020*; *Hug et al., 2021*; *Avrillon et al., 2024*). The manual editing consisted of (i) removing the spikes causing erroneous discharge rates (outliers), (ii) adding the discharge times clearly separated from the noise, (iii) recalculating the separation vector, (iv) reapplying the separation vector on the entire EMG signals, and (v) repeating this procedure until the selection of all the discharge times is achieved. The manual editing of potential missed discharge times and falsely identified discharge times was never immediately accepted. Instead, the procedure was consistently followed by the application of the updated motor unit separation vector on the entire EMG signals to generate a new motor unit pulse train. Then, the manual editing was only accepted when the silhouette value increased or stayed well above the threshold of 0.9 quantified with the silhouette value (*Negro et al., 2016a*). Only these motor units were retained for further analysis.

## Validation of the decomposition

EMG decomposition was validated using both simulation of EMG signals and two-source validation.

For the simulation, we generated the series of discharge times of a group of 150 motor units in an anatomical model entailing a cylindrical muscle volume with parallel fibres (see *Konstantin et al., 2020*, for a full description of the model), in which subcutaneous and skin layers separate the muscle from the surface electrodes. We set the radius of the muscle to 15 mm and the thicknesses of the subcutaneous and skin layers to 4 and 1 mm, respectively. The motor unit action potentials were detected in the model by a grid of 64 circular surface electrodes with a diameter of 1 mm arranged

in 5 columns and 13 rows (interelectrode distance: 4 mm). Following this, we estimated the rate of agreements between simulated and estimated series of motor unit discharge times, calculated as the ratio between correctly identified discharge times and the sum of correctly identified discharge times, missed discharge times, and falsely identified discharge times.

For the two-source validation, EMG signals were simultaneously recorded with an intramuscular linear array of 40 electrodes on a thin film and two grids of 64 surface electrodes. After the decomposition of both intramuscular and surface EMG signals, identified motor units were matched following the same procedure as for the elimination of duplicates. In short, the motor unit pulse trains were first aligned using a cross-correlation function. Two motor units identified with intramuscular and surface EMG decompositions were considered as matches when they had at least 30% of their discharge times in common. We calculated the accuracy of the decomposition using the rate of agreements between their series of discharge times.

## Proportion of the EMG signal represented by the decomposition

A synthetic EMG signal was reconstructed from the cumulative firing activity of the identified motor units. First, EMG signals from the 256 electrodes were differentiated in the column direction to obtain 236 single-differential EMG channels. Second, each differentiated EMG signal was segmented over successive windows of 25 ms centred around the discharge times of a motor unit. All the windows were averaged to estimate the average action potential waveform of that motor unit over the 236 each channel. The action potentials were then convolved with the series of discharge times to obtain trains of action potentials. These steps were repeated for all the identified motor units, and all the trains of action potentials were summed to reconstruct the synthetic EMG signals. At the end of this process, the ratio between the powers of synthetic and original EMG signals was calculated.

## Motor unit tracking

As explained in the section that described the decomposition framework and in previous studies (*Francic and Holobar, 2021*; *Škarabot et al., 2023a*; *Škarabot et al., 2023b*), a motor unit pulse train results from the projection of extended and whitened EMG signals onto a separation vector, optimised during a fixed-point algorithm. Discharge times are automatically identified from this motor unit pulse train, before visual inspection and manual editing.

In this study, motor units were tracked by slightly adapting this process. First, EMG signals from two successive contraction levels, say 10% and 20% MVC, were separately decomposed. The motor unit pulse trains, the identified discharge times, and the associated separation vectors were saved for each contraction level. Second, the EMG signals from the highest contraction level (20% MVC) were projected onto the separation vectors identified in the lowest contraction level (10% MVC). A new group of pulse trains was generated, which represented the firing activity of all the motor unit identified at 10% MVC, but during the contraction performed at 20% MVC. When the high peaks were clearly separated from the noise, the discharge times were automatically identified using the MATLAB functions *findpeaks.m* and *kmeans.m*. Third, these new series of discharge times were matched with those that have been initially identified at 20% MVC using the same approach as during two-source validation (>30% of common discharge times, see above). This process ended with three groups of motor units: (i) the motor units only identified at 10% MVC, (ii) the motor units identified at 10% and 20% MVC, and (iii) the motor units only identified at 20% MVC. These steps were repeated between all the successive contraction levels, i.e., from between 20% and 30% MVC to between 70% and 80% MVC. At the end of this iterative process, the following data were saved for each motor unit across the contraction levels where they were tracked: the instantaneous firing rates and force during ramp-up and ramp-down phases (*Figure 2*), the average firing rate during the plateau (*Figure 3*), the recruitment threshold (*Figure 1—figure supplement 2*).

## Motor unit uniqueness

The accuracy of motor unit tracking was further tested by confirming the uniqueness of each motor unit within the entire sample, assuming that each motor unit must have a unique representation of their action potentials across the array of surface electrodes (*Figure 1*; *Farina et al., 2008*). This was accomplished by calculating the RMSE between their action potentials across contractions and the action potentials of the rest of the motor units. EMG signals from the 256 electrodes were

differentiated in the column direction to obtain 236 single-differential EMG channels. For each motor unit, the single-differential EMG signals were segmented around the discharge times and averaged to identify an average action potential waveforms for each channel. The action potential waveforms were concatenated in a matrix of 236 rows (EMG channels) and 102 columns (time samples; 50 ms). This process was repeated for each contraction level where the motor unit was tracked. A first RMSE was calculated between the concatenated action potentials identified across contractions, and this value was defined as a reference. Then, RMSE was calculated between the concatenated action potentials of that motor unit and the concatenated action potentials of the rest of the motor units. A reference value lower than the 5th percentile of the distribution of RMSE calculated between motor units demonstrated that its distribution of action potentials across contractions were more similar than with the rest of the motor units, proving their uniqueness within the entire sample.

## Input-output function of motoneurons

The input-output function of each motor unit was characterised as the relation between its instantaneous firing rate and the muscle force. The instantaneous firing rates and muscle forces recorded during all the ramp-up phases where that motor unit was tracked were pooled. For example, the motor unit displayed in *Figure 2A* was tracked from 20% to 60% of the MVC. The relation between firing rates and force was characterised by comparing three curve fits:

- A linear fit: Firing rate (Force)=a(Force)+b
- A rising exponential: Firing rate (Force)=a(1 − e$^{(-Force/b)}$)+c
- A natural logarithm: Firing rate (Force)=a * ln(Force)+b

The parameters of the functions were estimated with the MATLAB functions *fittype.m* and *fit.m* using a non-linear least-squares solver with a maximum of 1000 iterations. The best model of the firing rate-force relation during the ramp-up phase was the fit with the lowest Bayesian information criterion (BIC). This criterion assesses the performance of a model by balancing its goodness of fit with its complexity (number of parameters). The BIC was calculated as follows:

$$\text{BIC} = n * \ln\left(sse / n\right) + p * \ln\left(n\right)$$

where n is the number of data samples on which the fit was estimated, sse is the sum of squares error, and p is the number of parameters used in the model.

The input-output functions of motor units during the ramp-down phase were characterised using the same approach, and we reported in the Results whether the best fit changed between the ramp-up and ramp-down phases.

## Rate coding of motor units

Most of motor units input-output functions were better characterised with a natural logarithm function. This function determines two stages in rate coding: a steep acceleration of firing rate followed by a slower linear increase in firing rate. Therefore, these two stages of the firing rate-force relation were further analysed. To estimate the acceleration of firing rate during the ramp-up phase, the first derivative of the firing rate-force relation fitted with a natural logarithm was computed as:

$$\text{Acceleration of firing rate}\left(\text{Force}\right) = a / \text{Force}$$

where a is the parameter of the equation of the natural logarithm function. The initial acceleration of firing rate was calculated as the value of the derivative at the recruitment threshold of the motor unit. The relation between the initial acceleration of firing rate and the recruitment threshold of all motor units across the pool was characterised for each muscle and each participant using non-linear least-squares solvers in MATLAB. The slow linear increase in firing rate that followed the initial steep acceleration of firing rate was characterised by comparing the average firing rate of motor units during the plateaus of force between successive contraction levels (*Figure 3*). Specifically, the rate of increase in firing rate for an increment of 10% MVC were calculated for each motor unit, and the relation between the rate of increase and the recruitment threshold of all motor units across the pool was characterised for each participant and each muscle using a linear fit in MATLAB. Pearson's r and p-values of these relations were reported in the Results.

### Hysteresis between recruitment and derecruitment thresholds

The relation between recruitment and derecruitment thresholds of all motor units across the pool was characterised for each participant and each muscle using non-linear and linear fits. As for the relations between the firing rate and force of each motor unit, the best fit was selected using the BIC. The differences between the recruitment and derecruitment thresholds (hysteresis) were calculated for each motor unit, with negative values indicating a hysteresis and a positive value indicating a reverse hysteresis.

### Statistical analyses

All statistical analyses were performed with RStudio (USA). Quantile-quantile plots and histograms were displayed to check the normality of the data distribution. If the distribution was determined not to be normal, they were transformed to remove the skew. All statistical analyses were performed using linear mixed effect models implemented in the R package *lmerTest* with the Kenward-Roger method to estimate the denominator degrees of freedom and the p-values. This method considers the dependence of data points (i.e. individual motor unit) in each participant. When necessary, multiple comparisons were performed using the R package *emmeans*, which adjusts the p-value using the Tukey or the Bonferroni method. The significance level was set at 0.05. Values are reported as mean ± standard deviation.

Initial accelerations of firing rate and rates of increase in firing rate between successive contraction levels were compared between motor units using a linear mixed effect model, with *muscle* (TA; VL) and *threshold* (low; medium; high) as fixed effects and *participant* as a random effect.

The difference between motor units' recruitment and derecruitment thresholds were compared using a linear mixed effect model, with *muscle* (TA; VL) and *threshold* (low; medium; high) as fixed effects and *participant* as a random effect. Whether the difference between the two thresholds were significantly different from 0 (absence of hysteresis) was further tested using function *contrast* in the package *emmeans*.

## Acknowledgements

Simon Avrillon is supported by the French National Research Agency through Nantes Excellence Trajectory (NExT, ANR-16-IDEX-0007). Simon Avrillon and Dario Farina are supported by the BBSRC, 'Neural Commands for Fast Movements in the Primate Motor System' (BB/V00896X). François Hug is supported by the UCAJEDI Investments in the Future project managed by the French National Research Agency (ANR-15-IDEX-01) and the French National Research Agency (Neuromotor, ANR-24-CE17-5805). Dario Farina is supported by the European Research Council Synergy Grant NaturalBionicS (contract #810346) and the EPSRC Transformative Healthcare, NISNEM Technology (EP/T020970). Roger M Enoka is partially supported by an award from the National Multiple Sclerosis Society in the USA (project RG-2206-39688).

## Additional information

### Funding

| Funder | Grant reference number | Author |
|---|---|---|
| Agence Nationale de la Recherche | ANR-16-IDEX-0007 | Simon Avrillon |
| Biotechnology and Biological Sciences Research Council | BB/V00896X | Simon Avrillon Dario Farina |
| Engineering and Physical Sciences Research Council | EP/T020970 | Dario Farina |
| Agence Nationale de la Recherche | ANR-24-CE17-5805 | François Hug |

| Funder | Grant reference number | Author |
|---|---|---|
| European Research Council | #810346 | Dario Farina |
| Future project managed by the French National Research Agency | ANR-15-IDEX-01 | François Hug |
| National Multiple Sclerosis Society | RG-2206-39688 | Roger M Enoka |

The funders had no role in study design, data collection and interpretation, or the decision to submit the work for publication.

### Author contributions

Simon Avrillon, Conceptualization, Data curation, Software, Formal analysis, Validation, Investigation, Visualization, Methodology, Writing – original draft, Writing – review and editing; François Hug, Resources, Data curation, Investigation, Methodology, Writing – review and editing; Roger M Enoka, Resources, Writing – review and editing; Arnault HD Caillet, Conceptualization, Data curation, Investigation, Methodology, Writing – review and editing; Dario Farina, Conceptualization, Resources, Supervision, Writing – review and editing

### Author ORCIDs

Simon Avrillon  https://orcid.org/0000-0002-2226-3528
Arnault HD Caillet  https://orcid.org/0000-0001-6146-1829
Dario Farina  https://orcid.org/0000-0002-7883-2697

### Ethics

Ethical committees approved the study (quadriceps protocols: Comité; de Protection des Personnes Ouest, no. 23.00453.000166; tibialis anterior protocol: Imperial College London, no. 18IC4685). All participants provided their informed written consent before the beginning of the experiment.

Reviewer #1 (Public review): https://doi.org/10.7554/eLife.97085.3.sa1
Reviewer #2 (Public review): https://doi.org/10.7554/eLife.97085.3.sa2
Reviewer #3 (Public review): https://doi.org/10.7554/eLife.97085.3.sa3
Author response https://doi.org/10.7554/eLife.97085.3.sa4

## Additional files

### Supplementary files
• MDAR checklist

### Data availability

All data recorded for this study are available on Figshare. Code and apps associated with this study are available on Github at https://github.com/simonavrillon/MUdictionnary (copy archived at *Avrillon, 2023*).

The following dataset was generated:

| Author(s) | Year | Dataset title | Dataset URL | Database and Identifier |
|---|---|---|---|---|
| Avrillon S, Hug F, Caillet A, Farina D | 2024 | Data for the preprint 'The decoding of extensive samples of motor units in human muscles reveals the rate coding of entire motoneuron pools' | https://doi.org/10.6084/m9.figshare.24640944 | figshare, 10.6084/m9.figshare.24640944 |

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
