## [Editor Report · eLife Assessment]

Leveraging state-of-the-art experimental and analytical approaches, this **important** study characterizes the recruitment and activation of large populations of human motor units during slow isometric contractions in two lower limb muscles. Evidence for the main claims is **solid** and advances our understanding of how humans generate and control voluntary force.

---

## [Referee Report · Reviewer #1 (Public review)]

Summary:

The Avrillon et al. explore the neural control of muscle by decomposing the firing activity of constituent motor units from the grid of surface electromyography (EMG) in the Tibialis (TA) Anterior and Vastus Lateralis (VL) during isometric contractions. The study involves extensive samples of motor units across the broadest range of voluntary contraction intensities up to 80% of MVC. The authors examine rate coding of the population of motor units, which describes the instantaneous firing rate of each motor unit as a function of muscle force. This relationship is characterized by a natural logarithm function that delineates two distinct phases: an initial phase with a steep acceleration in firing rate, particularly pronounced in low-threshold motor units, and a subsequent modest linear increase in firing rate, more significant in high-threshold motor units.

Strengths:

The study makes a significant contribution to the field of neuromuscular physiology by providing a detailed analysis of motor unit behavior during muscle contractions in a few ways.

(1) The significance lies in its comprehensive framework of motor unit activity during isometric contractions in the broad range of intensities, providing insights into the non-linear relationship between the firing rate and the muscle force. The extensive sample of motor units across the pool confirms the observation in animal studies in which the the spinal motoneuron exhibits a discharge consists of the distinct phases in response to synaptic currents, under the influence of persistent inward currents. As such, it is now reasonable to state the human motor units across the pool are also under control of gain modulation via some neuromodulatory effects in addition to synaptic inputs arising from ionotropic effects.

(2) The firing scheme across in the entire motoneuron pool revealed in this study reconciles the discrepancy in firing organization under debate; i.e., whether it is 'onion skin' like or not (Heckman and Enoka 2012). The onion skin like model states that the low threshold motor units discharge higher than high threshold motor units and has been held for long time because the firing behaviors were examined in a partial range of contraction force range due to technical limitations. This reconciliation is crucial because it is fundamental to modelling the organization of motor unit recruitment and rate coding to achieve a desired force generation to advance our understanding of motor control.

(3) The extensive data collection with a novel blind source separation algorithm on the expanded number of channel of surface EMG signal provides a robust dataset that enhances the reliability and validity of findings, setting a new standard for empirical studies in the field. \par

Collectively, this study fills several knowledge gaps in the field and advances our understanding the mechanism underlying the isometric force generation.

---

## [Referee Report · Reviewer #2 (Public review)]

Avrillon et al. provides a comprehensive assessment of firing rate parameters from a large percentage of the motor unit pool, in two muscles, during voluntary isometric contractions. The authors have used new quantitative methods to extract more unique motor units across contractions than prior studies. This was achieved by recording muscle fibre action potentials from four high density surface electromyogram (HDsEMG) arrays, quantifying residual EMG comparing the recorded and data-based simulation (Fig. 1A-B), and developing a metric to compare the spatial identification for each motor unit (Fig. 1D-E). From identified motor units, the authors have provided a detailed characterization of recruitment and firing rate responses during slow voluntary isometric contractions in the vastus lateralis and tibialis anterior muscles up to 75-80% of maximum intensity. In the lower limb it is interesting how lower threshold motor units have firing rate responses that saturate, whereas higher threshold units that presumably produce higher muscle contractile forces continue to increase their firing rate. Conceptually, the authors rightly focus on the literature of intrinsic motoneurone properties, but in vivo, other possibilities (that are difficult to measure in awake human participants) are that the form of descending supraspinal drive, spinal network dynamics and afferent inputs may have different effects across motor unit sizes, muscles and types of contractions. These results from single trail contractions and with a larger sample of motor units, supports the summary rate coding profiles of motor units in the extensor digitorum communis muscle (Monster and Chan, 1977).

---

## [Referee Report · Reviewer #3 (Public review)]

Summary:

This is an interesting manuscript which uses state of the art experimental and simulation approaches to quantify motor unit discharge patterns in the human TA and VL. The non-linear profiles of motor unit discharge were calculated and found to have an initial acceleration phase followed by an attenuation phase. Lower threshold motor units had a larger gain of the initial acceleration whereas the higher threshold motor unit had a higher gain in the attenuation phase. These data represent a technical feat and are important for understanding how humans generate and control voluntary force.

Strengths:

The authors used rigorous, state-of-the art analyses to decompose and validate their motor unit data during a wide range of voluntary efforts.

Analyses are clearly presented, applied, and visualized.

The supplemental data provides important transparency.

Weaknesses:

Number of participants and muscles tested are relatively small - particularly given the constraints on yield. It is unclear if this will translate to other motor pools. The justification for TA and VL should be provided.

While in impressive effort was made to identify and track motor units across a range of contractions, it appears that a substantial portion of muscle force was not identified. Though high intensity contractions are challenging to decompose - the authors are commended in their technical ability in recording population motor unit discharge times with recruitment thresholds up to 75% a participant's maximal voluntary contractions. However previous groups have seen substantial recruitment motor units above 80% and even 90% maximum activation in the soleus. Given the innervation ratios of higher threshold motor units, if recruitment continued to 100%, the top quartile would likely represent a substantial portion of the traditional fast-fatigable motor units. It would be highly interesting to understand the recruitment and rate coding of the highest threshold motor units, at a minimum I would suggest using terms other than "entire range" or "full spectrum of recruitment thresholds"

The quantification of hysteresis using torque appears to make self-evident the observation that lower threshold motor units demonstrate less hysteresis with respect to torque - If there was motor unit discharge there will be force. I believe this limitation goes beyond the floor effects discussed in the manuscript. Traditionally individuals have used the discharge of a lower threshold unit as the measure on which to apply hysteresis analyses to infer ion channel function in human spinal motoneurons.

The main findings are not entirely novel. See Monster and Chan 1977 and Kanosue et al 1979

Comments on revisions:

I thank the authors for their thoughtful revision.

Just to confirm, the ranges for motor unit yield are for a single contraction. So, for example, in a participant there were 71 unique and concurrently active VL motor units able to be decomposed.

---

## [Author Response]

The following is the authors’ response to the original reviews.

**Reviewer #1 (Public Review):**
Summary:This study explores the neural control of muscle by decomposing the firing activity of constituent motor units from the grid of surface electromyography (EMG) in the Tibialis (TA) Anterior and Vastus Lateralis (VL) during isometric contractions. The study involves extensive samples of motor units across the broadest range of voluntary contraction intensities up to 80% of MVC. The authors examine the rate coding of the population of motor units, which describes the instantaneous firing rate of each motor unit as a function of muscle force. This relationship is characterized by a natural logarithm function that delineates two distinct phases: an initial phase with a steep acceleration in firing rate, particularly pronounced in low-threshold motor units, and a subsequent modest linear increase in firing rate, more significant in high-threshold motor units.Strengths:The study makes a significant contribution to the field of neuromuscular physiology by providing a detailed analysis of motor unit behavior during muscle contractions in a few ways.(1) The significance lies in its comprehensive framework of motor unit activity during isometric contractions in a broad range of intensities, providing insights into the non-linear relationship between the firing rate and the muscle force. The extensive sample of motor units across the pool confirms the observation in animal studies in which the spinal motoneuron exhibits a discharge consisting of distinct phases in response to synaptic currents, under the influence of persistent inward currents. As such, it is now reasonable to state the human motor units across the pool are also under the control of gain modulation via some neuromodulatory effects in addition to synaptic inputs arising from ionotropic effects.(2) The firing scheme across the entire motoneuron pool revealed in this study reconciles the discrepancy in firing organization under debate; i.e., whether it is 'onion skin' like or not (Heckman and Enoka 2012). The onion skin like model states that the low threshold motor units discharge higher than high threshold motor units and have been held for a long time because the firing behaviors were examined in a partial range of contraction force range due to technical limitations. This reconciliation is crucial because it is fundamental to modelling the organization of motor unit recruitment and rate coding to achieve a desired force generation to advance our understanding of motor control.(3) The extensive data collection with a novel blind source separation algorithm on the expanded number of channels of surface EMG signal provides a robust dataset that enhances the reliability and validity of findings, setting a new standard for empirical studies in the field.Collectively, this study fills several knowledge gaps in the field and advances our understanding of the mechanism underlying the isometric force generation.

We thank the reviewer for their positive appreciation of our work.

Weaknesses:Although the findings and claims based on them are mostly well aligned, some accounts of the methods and claims need to be clarified.(1) The authors examine the input-output function of a motor unit by constructing models, using force as an input and discharge rate as an output. It sounds circular, or the other way around to use the muscle force as an input variable, because the muscle force is the result of motor unit discharges, not the cause that elicits the discharges. More specifically, as a result of non-linear interactions of synchronous and/or asynchronous discharges of a population of a given motoneuron pool that give rise to transient increase/maintenance in twitch force, the gross muscle force is attained. I acknowledge that it is extremely challenging experimentally to measure synaptic currents impinging upon the spinal motoneurons in human subjects and the author has an assumption that the force could be used as a proxy of synaptic currents. However, it is necessary to explicitly provide the caveats and rationale behind that. Force could be used as the input variable for modelling.

Force is indeed used in this study as a proxy of the common excitatory synaptic currents as their direct measurement is not possible in vivo in humans. It is worth noting that this approach has been extensively used in the past by many groups to study rate coding (e.g., Monsters & Chan, De Luca’s, Heckman’s, and Fuglevand’s groups). Heckman’s, Gorassini’s, Fuglevand’s groups and others have considered the non-linearities in the relation between motor unit firing rates and muscle force in humans as an indicator of the impact of neuromodulation on motor unit behaviour and changes of the intrinsic properties of motoneurons.

One could also use the cumulative spike train as a more direct estimate of common excitatory inputs, assuming that it is possible to identify a group of motor units not influenced by PICs, as done when selecting a reference low-threshold motor neuron in the delta F method (Gorassini et al., 1998), or the cumulative spike train of low-threshold motor neurons (Afsharipour et al., 2020). However, this approach was not possible in our study as we did not have the same units across contractions to estimate cumulative spike trains. It was therefore not possible to pool the data across contractions as we did to generate force/firing rate relations on the widest range of force.

We added a sentence in the discussion to highlight this limitation (P19, L470):

‘This result must be confirmed with a more direct proxy of the net synaptic drive, such as the firing rate of a reference low-threshold motor neuron used in the delta F method (Gorassini et al., 1998), or the cumulative spike train of low-threshold motor neurons (Afsharipour et al., 2020)’.

(2) The authors examine the firing organizations in TA and VL in this study without explicit purposes and rationale for choosing these muscles. The lack of accounts makes it hard for the readers to interpret the data presented, particularly in terms of comparing the results from the different muscles.

We wanted to compare the rate coding of pools of motor units from proximal (VL) and distal (TA) muscles within the lower limb. Indeed, distal and proximal muscles exhibit differences in rate coding and spatial recruitments (De Luca et al., 1982, J Physiol), potentially due to different levels of recurrent inhibition (Cullheim & Kellerth, 1978, J Physiol; Rossi & Mazzocchio, 1991, Exp Brain Res; Edgley et al., 2021, J Neurosci) or different levels of neuromodulation depending on their involvement (or not) in postural control (Hoonsgaard et al., 1988, J Physiol; Kim et al., 2020, J Neurophysiol).

We added a paragraph at the beginning of the result section to support our muscle choice (P6; L137): ‘16 participants performed either isometric dorsiflexion (n = 8) or knee extension tasks (n = 8) while we recorded the EMG activity of the tibialis anterior (TA - dorsiflexion) or the vastus lateralis (VL – knee extension) with four arrays of 64 surface electrodes (256 electrodes per muscle). The motoneuron pools of these two muscles of the lower limb receive a large part of common input (Laine et al., 2015; Negro et al., 2016a), constraining the recruitment of their motor units in a fixed order across tasks. They are therefore good candidates for an accurate description of rate coding. Moreover, we wanted to determine whether differences in rate coding observed between proximal and distal muscles in the upper limb (De Luca et al., 1982) were also present in the lower limb.’.

Another factor that guided our muscle choice was the low risk of crosstalk. For this, we verified with ultrasound that our arrays of 256 electrodes only covered the muscle of interest, staying away from the neighbouring muscles. This was possible as superficial muscles from the leg are bulkier than those from the upper limb. Given the small diameter of each electrode (2 mm), it is unlikely that the motor units from the neighbouring muscles were in the recorded muscle volume (Farina et al., 2003, IEEE Trans Biomed Eng)

(3) In the methods, the author described the manual curation process after applying the blind source separation algorithm. For the readers to understand the whole process of decomposition and to secure rigor and robustness of the analyses, it would be necessary to provide details on what exact curation is performed with what criteria.

The manual curation of EMG decomposition with blind source separation is different from what is classically done with intramuscular EMG and template-matching algorithms.

In short, our decomposition algorithm uses fast independent component analysis (fastICA) to retrieve motor unit spike trains from the EMG signals. For this, it iteratively optimises a set of weights, i.e., a separation vector, for each motor unit. The projection of the EMG signals on this separation vector generates a sparse motor unit pulse train, with most of its samples close to zero and only a few samples close to one (Figure 1B). The discharge times are estimated from this motor unit pulse train using a peak detection function and a k-mean classification with two classes to separate the high peaks (spikes) from the low peaks (noise and other motor units).

The manual curation consists of inspecting the automatic detection of the peaks of the motor unit pulse train and manually add missed peaks (missed discharge times) or remove wrongly detected peaks. Then, the separation vector is updated using the correct discharge times and the motor unit pulse train recalculated. This procedure generally improves the distance between the discharge times and the noise, which confirm the accuracy of the manual curation. If that’s not the case, the motor unit is discarded from the analyses.

We added a section on manual editing in the methods (P23, L615):

‘At the end of these automatic steps, all the motor unit pulse trains and identified discharge times were visually inspected, and manual editing was performed to correct the false identification of artifacts or the missed discharge times (Del Vecchio et al., 2020; Hug et al., 2021; Avrillon et al., 2023). The manual editing consisted of (i) removing the spikes causing erroneous discharge rates (outliers), (ii) adding the discharge times clearly separated from the noise, (iii) recalculating the separation vector, (iv) reapplying the separation vector on the entire EMG signals, and (v) repeating this procedure until the selection of all the discharge times is achieved. The manual editing of potential missed discharge times and falsely identified discharge times was never immediately accepted. Instead, the procedure was consistently followed by the application of the updated motor unit separation vector on the entire EMG signals to generate a new motor unit pulse train. Then, the manual editing was only accepted when the silhouette value increased or stayed well above the threshold of 0.9 quantified with the silhouette value (Negro et al., 2016b). Only these motor units were retained for further analysis.’

(4) In Figure 3, the early recruited units tend to become untraceable in the higher range of contraction. This is more pronounced in the muscle VL. This limitation would ambiguate the whole firing curve along the force axis and therefore limitation and the applicability in the different muscles needs to be discussed.

The loss of low threshold motor units in the higher range of contractions was caused either by the decrease in signal-to-noise ratio for small motor units when many larger ones are recruited, or by the cancellation of the surface action potentials of the small units in the interference electromyographic signal, or by the recruitment of a motor unit with a very similar spatio-temporal filter (an example is shown in the figure below). In the latter case, the motor unit pulse train contains peaks that represent the discharge times of both motor units (green and red dots in the simulated example below), making them undistinguishable by the operator during manual editing.

**Author response image 1. sa4fig1:** 

This was discussed in the results (P7; L190):

‘On average, we tracked 67.1 ± 10.0% (25th–75th percentile: 53.9 – 80.1%) of the motor units between consecutive contraction levels (10% increments, e.g., between 10% and 20% MVC) for TA and 57.2 ± 5.1% (25th–75th percentile: 46.6 – 68.3%) of the motor units for VL (Figure S2). There are two explanations for the inability to track all motor units across consecutive contraction levels. First, some motor units are recruited at higher targets only. Second, it is challenging to track small motor units beyond a few contraction levels due to a lower signal-to-noise ratio for the small motor units when larger motor units are recruited, or signal cancellation (Keenan et al., 2005; Farina et al., 2014a).’

However, we believe that it had a limited impact on the output of the paper, as the non-linear portion of the rate coding/force relation due to the persistent inward currents occurs during the first seconds after recruitment, before plateauing (for a review see Binder et al., 2020, Physiology).

(5) It is unclear how commonly the notion "the long-held belief that rate coding is similar across motor units from the same pool" is held among the community without a reference. Different firing organizations have been modelled and discussed in the seminal paper by Fuglevand et al. (1993) and as far as I understand, the debate has not converged to a specific consensus. As such, any reference would be required to support the claim the notion is widely recognized.

In the paper of Fuglevand et al., (1993, J Neurophysiol), all the motor units had the same rate coding pattern relative to the excitatory input, though they changed the slope of the relations and the saturation threshold of motor units between simulations. This is similar to the paper of De Luca & Contessa (2012, J Neurophysiol), where the equation used to simulate the rate coding was non-linear, but consistent across motor units.

We added these citations to the text:

‘Overall, we found that motor units within a pool exhibit distinct rate coding with changes in force level (Figure 2 and 3), which contrasts with the long-held belief that rate coding is similar across motor units from the same pool (Fuglevand et al., 1993; De Luca and Contessa, 2012).’

(6) The authors claim that the firing behavior as a function of force is well characterized by a natural logarithmic function, which consists of initial steep acceleration followed by a modest increase in firing rate. Arguably the gain modulation in firing rate could be attributed to a neuromodulatory effect on the spinal motoneuron, which has been suggested by a number of animal studies. However, the complexity of the interactions between ionotropic and neuromodulatory inputs to motoneurons may require further elucidation to fully understand the mechanisms of neural control; it is possible to consider the differential acceleration among different threshold motor units as a differential combinatory effect of ionotropic and neuromodulatory inputs, but it is not trivially determined how differentially or systematically the inputs are organized. Likewise, the authors make an account for the difference in firing rate between TA and VL in terms of different amounts or balances of excitatory and inhibitory inputs to the motoneuron pool, but again this could be explained by other factors, such as a different extent of neuromodulatory effects. To determine the complexity of the interactions, further studies will be warranted.

We appreciate the reviewer’s view on this point, as we indeed only indirectly inferred the combination of neuromodulatory and ionotropic inputs to motoneurons in this study. A more direct manipulation of the sources of neuromodulatory and ionotropic inputs will be required in the future to directly highlight the mechanisms responsible for these variations in rate coding within pools. However, it is also worth noting that the acceleration in firing rate, the increase in firing rate during the ramp up, and the hysteresis between ramps up and downs have been used to infer the distribution of ionotropic and neuromodulatory inputs from the firing rate/force relations (Johnson et al., 2017; Beauchamp et al., 2023; Chardon et al., 2023). This approach has been validated with hundreds of thousands of simulations using a biophysical model of motor neurons (Chardon et al., 2023). There is also a series of studies in humans showing how the absence of neuromodulation modulated via inhibitory inputs (Revill & Fuglevand, 2017) or medication blocking serotonin receptors (Goodlich et al., 2023) impact the non-linearity of the firing rate/force relation. Therefore, we are confident that the differences observed within and between pools are linked to different distribution of excitatory/inhibitory inputs and neuromodulation.

We added a sentence in the discussion to highlight this point (P18; L435):

‘Taken together, these results show how ionotropic and neuromodulatory inputs to motoneurons uniquely combine to generate distinct rate coding across the pool, even if a more direct manipulation of the sources of neuromodulatory and ionotropic inputs will be required to directly estimate their interactions.’

(7) It is unclear with the account " ... the bandwidth of muscle force is < 10Hz during isometric contraction" in the manuscript alone, and therefore, it is difficult to understand the following claim. It appears very interesting and crucial for motor unit discharge and force generation and maintenance because it would pose a question of why the discharge rate of most motor units is higher than 10Hz, despite the bandwidth being so limited, but needs to be elaborated.

We described the slow fluctuations in smoothed firing rates associated with the variations in force observed during isometric contractions. The bandwidth of muscle force is lower than 10Hz due to the contractile properties of muscle tissues (Baldissera et al., 1998, J Physiol). Having an average firing rate higher than this bandwidth enables the pool of motor neurons to effectively transmit the common inputs (the main discriminant of muscle force) over this bandwidth without distortion (Farina et al., 2014, J Physiol). Increasing the firing rate beyond the muscle bandwidth also increases the power of the spike train at the direct current frequency (frequency equal to 0) since this power is related to the number of spikes per second. Thus, increasing the firing rate well beyond the muscle bandwidth still has a clear effect in force. To illustrate this point, note that electrical stimuli delivered at 100 Hz can lead to an increase in muscle force.

**Reviewer #2 (Public Review):**
Summary:The motivation for this study is to provide a comprehensive assessment of motor unit firing rate responses of entire pools during isometric contractions. The authors have used new quantitative methods to extract more unique motor units across contractions than prior studies. This was achieved by recording muscle fibre action potentials from four high-density surface electromyogram (HDsEMG) arrays (Caillet et al., 2023), quantifying residual EMG comparing the recorded and data-based simulation (Figure 1A-B), and developing a metric to compare the spatial identification for each motor unit (Figure 1D-E). From identified motor units, the authors have provided a detailed characterization of recruitment and firing rate responses during slow voluntary isometric contractions in the vastus lateralis and tibialis anterior muscles up to 80% of maximum intensity. In the lower limb, it is interesting how lower threshold motor units have firing rate responses that saturate, whereas higher threshold units that presumably produce higher muscle contractile forces continue to increase their firing rate. In many ways, these results agree with the rate coding of motor units in the extensor digitorum communis muscle (Monster and Chan, 1977). The paper is detailed, and the analyses are well explained. However, there are several points that I think should be addressed to strengthen the paper.

We thank the reviewer for their positive appreciation of our work.

General comments:(1) The authors claim they have measured the complete rate coding profiles of motor units in the vastus lateralis and tibialis anterior muscles. However, this study quantified rate coding during slow and prolonged voluntary isometric contractions whereas the function of rate coding during movements (Grimby and Hannerz, 1977) or more complex isometric contractions (Cutsem and Duchateau, 2005; Marshall et al., 2022) remains unexplored. For example, supraspinal inputs may not scale the same way across low and higher threshold motor units, or between muscles (Devanne et al., 1997), making the response of firing rates to increasing isometric contraction force less clear.

We agree with the reviewer that rate coding strategies may vary with the velocity and the type of contractions (Duchateau & Enoka, 2008, J Physiol). It is thus likely that the firing rate would increase during the first milliseconds of fast contractions, with the occurrence of doublets (Cutsem and Duchateau, 2005, J Physiol; Del Vecchio et al., 2019, J Physiol), or that motor unit firing rate may be lower during lengthening than shortening contractions (Duchateau & Enoka, J Physiol).

However, the decomposition of EMG signals in non-stationary conditions remains challenging, and is still limited to slow varying patterns of force (Chen et al., 2000, Oliveira & Negro, 2021, Mendez Guerra et al., 2024, Yeung et al., 2024). Future methodological developments will be required to expand our findings to other patterns of force.

Conceptually, the authors focus on the literature on intrinsic motoneurone properties, but in vivo, other possibilities are that descending supraspinal drive, spinal network dynamics, and afferent inputs have different effects across motor unit sizes, muscles, and types of contractions. Also, the influence from local muscles that act as synergists (e.g., vastii muscles for the vastus lateralis, and peroneal muscles that evert the foot for the tibialis anterior) or antagonists (coactivation during higher contraction intensities would stiffen the joint) may provide differential forms of proprioceptive feedback across motor pools.

The reviewer is right that differences in spinal network dynamics and afferent inputs may explain the differences in rate coding observed between the two muscles. Indeed, computational models have shown how the pattern of inhibitory inputs may affect the increase in firing rate during linear increase in force (Powers & Heckman, 2017, J Neurophysiol; Chardon et al., 2023, Elife). Specifically, the difference observed between proportional inhibitory inputs vs. a push pull pattern mirror the differences observed here between the TA (push-pull like pattern) and the VL (proportional pattern). This difference may reflect the impact of various pathways of inhibition, such as reciprocal inhibition or recurrent inhibition from homonymous motor units or motor units from synergistic muscles.

These points have been further discussed in the manuscript (P19; L475):

‘The increase in firing rate was also significantly greater for TA motor units than for those in VL. This difference may reflect a varying balance between excitatory/inhibitory synaptic inputs and neuromodulation due to multiple spinal circuits (Heckman and Binder, 1993; Heckman et al., 2008; Johnson et al., 2017; Powers and Heckman, 2017; Chardon et al., 2023; [77]). Specifically, the strength of recurrent and reciprocal inhibitory inputs to motoneurons innervating VL and TA, and their proportional or inverse covariation with excitatory inputs, respectively, may explain the differences in rate limiting and maximal firing rates (Heckman and Binder, 1993; Heckman et al., 2008; Johnson et al., 2017; Powers and Heckman, 2017; Chardon et al., 2023; Škarabot et al., 2023). Thus, the motor units from the VL may receive more recurrent inhibition than those of distal muscles, though direct evidence of these differences remains to be found in humans (Windhorst, 1996). Interestingly, similar differences in rate coding were previously observed between proximal and distal muscles of the upper limb (De Luca et al., 1982). However, other muscles that serve different functions within the human body, such as muscles from the face, have different rate coding characteristics with much higher firing rates (Kirk et al., 2021). Future work should investigate those muscles and other to reveal the myriads of rate coding strategies in human muscles.’

(2) The evidence that the entire motor unit pool was recorded per muscle is not clear. There appears to be substantial residual EMG (Figure 1B), signal cancellation of smaller motor units (lines 172-176), some participants had fewer than 20 identified motor units, and contractions never went above 80% of MVC. Also, to my understanding, there remains no gold-standard in awake humans to estimate the total motor unit number in order to determine if the entire pool was decomposed.

The reviewer is right that we did not decode the full pool of motor units. As indicated in the initial version of the manuscript (e.g. title, introduction), we considered that we identified an extensive sample of motor units representative of the dynamic of the pool. This claim was supported by the identification of motor units with recruitment thresholds ranging from 0 to 75% of the maximal force.

This statement was in the introduction (P4; L109): ‘We were able to identify up to ~200 unique active motor units per muscle and per participant in two human muscles in vivo, yielding extensive samples of motor units that are representative of the entire motoneuron pools (Caillet et al., 2023a).’

Furthermore, using four HDsEMG arrays also raises questions about how some channels were placed over non-target muscles, and if motor units were decomposed from surrounding synergists.

A factor that guided our muscle choice was the low risk of crosstalk. For this, we verified with ultrasound that our arrays of 256 electrodes only covered the muscle of interest, staying away from the neighbouring muscles. This was possible as superficial muscles from the leg are bulkier than those from the upper limb. Given the small diameter of each electrode (2 mm), it is unlikely that the motor units from the neighbouring muscles were in the recorded muscle volume.

(3) The authors claim (Abstract L51; Discussion L376) that a commonly held view in the field is that rate coding is similar across motor units from the same pool. Perhaps this is in reference to some studies that have carefully assessed lower threshold motor units during lower force ramp contractions (e.g., Fuglevand et al., 2015; Revill and Fuglevand, 2017). However, a more complete integration of the literature exploring motor unit firing rate responses during rapid isometric contractions, comparing different muscles and contraction intensities would be helpful. From Figure 3, the range of rate coding in the tibialis anterior (~7-40 Hz) is greater than the vastus lateralis (~5-22 Hz) muscle across contraction levels. In agreement with other studies, the range of rate coding within some muscles is different than others (Kirk et al., 2021) and during maximal intensity (Bellemare et al., 1983) or rapid contractions (Desmedt and Godaux, 1978). Likewise, within a motor pool, there is a diversity of firing rate responses across motor units of different sizes as a function of isometric force (Monster and Chan, 1977; Desmedt and Godaux, 1977; Kukula and Clamann, 1981; Del Vecchio et al., 2019; Marshall et al., 2022). A strength of this paper is how firing rate responses are quantified across a wide range of motor unit recruitment thresholds and between two muscles. I suggest improving clarity for the general reader, especially in the motivation for testing two lower limb muscles, and elaborating on some of the functional implications.

We thank the reviewer for his input on this question. We have added references to these works and lines of research in the discussion:

(P18; L449): ‘In addition, rate coding patterns should also vary with the pattern of contractions, with fast contractions lowering the range of recruitment thresholds within motoneuron pools (Desmedt and Godaux, 1977b, 1979; van Bolhuis et al., 1997). The variability in rate coding observed here between motor units from the same pool could lead to small deviations from the size principle sometimes observed between pairs of units during isometric contractions with various patterns of force (Desmedt and Godaux, 1979; Marshall et al., 2022) or during the derecruitment phase (Bracklein et al., 2022).’ (P19; L487): ‘However, other muscles that serve different functions within the human body, such as muscles from the face, have different rate coding characteristics with much higher firing rates (Kirk et al., 2021). Future work should investigate those muscles and other to reveal the myriads of rate coding strategies in human muscles.’

In addition to the responses above, we have added a section at the beginning of the results to motivate the choice of the muscles (P6; L137):

‘16 participants performed either isometric dorsiflexion (n = 8) or knee extension tasks (n = 8) while we recorded the EMG activity of the tibialis anterior (TA - dorsiflexion) or the vastus lateralis (VL – knee extension) with four arrays of 64 surface electrodes (256 electrodes per muscle). The motoneuron pools of these two muscles of the lower limb receive a large part of common input (Laine et al., 2015; Negro et al., 2016a), constraining the recruitment of their motor units in a fixed order across tasks. They are therefore good candidates for an accurate description of rate coding. Moreover, we wanted to determine whether differences in rate coding observed between proximal and distal muscles in the upper limb (De Luca et al., 1982) were also present in the lower limb.’.

**Reviewer #3 (Public Review):**
Summary:This is an interesting manuscript that uses state-of-the-art experimental and simulation approaches to quantify motor unit discharge patterns in the human TA and VL. The non-linear profiles of motor unit discharge were calculated and found to have an initial acceleration phase followed by an attenuation phase. Lower threshold motor units had a larger gain of the initial acceleration whereas the higher threshold motor unit had a higher gain in the attenuation phase. These data represent a technical feat and are important for understanding how humans generate and control voluntary force.Strengths:The authors used rigorous, state-of-the-art analyses to decompose and validate their motor unit data during a wide range of voluntary efforts.The analyses are clearly presented, applied, and visualized.The supplemental data provides important transparency.

We thank the reviewer for their positive appreciation of our work.

Weaknesses:The number of participants and muscles tested are quite small - particularly given the constraints on yield. It is unclear if this will translate to other motor pools. The justification for TA and VL should be provided.

One strength of our study is to provide relations between key-parameters of rate coding (acceleration in firing rate, increase in firing rate, hysteresis) and the recruitment thresholds of motor units within two different pools, and for each individual participant. These relations were consistent across all the participants (Figures 2 to 4), making us confident that increasing the sample size would not change the conclusions of the study.

It is likely that the differences observed here between the VL and TA will also appear between other muscles of the leg, due to differences in the arrays of excitatory and inhibitory inputs they receive, the pattern of inhibitory inputs during increases in force (recurrent/reciprocal inhibition), and different levels of neuromodulation (Johnson et al., 2017, J Neurophysiol; Beauchamp et al., 2023; J Neural Eng). We have added a paragraph in the results to motivate our choice of muscles (P6; L137):

‘16 participants performed either isometric dorsiflexion (n = 8) or knee extension tasks (n = 8) while we recorded the EMG activity of the tibialis anterior (TA - dorsiflexion) or the vastus lateralis (VL – knee extension) with four arrays of 64 surface electrodes (256 electrodes per muscle). The motoneuron pools of these two muscles of the lower limb receive a large part of common input (Laine et al., 2015; Negro et al., 2016a), constraining the recruitment of their motor units in a fixed order across tasks. They are therefore good candidates for an accurate description of rate coding. Moreover, we wanted to determine whether differences in rate coding observed between proximal and distal muscles in the upper limb (De Luca et al., 1982) were also present in the lower limb.’.

While an impressive effort was made to identify and track motor units across a range of contractions, it appears that a substantial portion of muscle force was not identified. Though high-intensity contractions are challenging to decompose - the authors are commended for their technical ability to record population motor unit discharge times with recruitment thresholds up to 75% of a participant's maximal voluntary contractions. However previous groups have seen substantial recruitment of motor units above 80% and even 90% maximum activation in the soleus. Given the innervation ratios of higher threshold motor units, if recruitment continued to 100%, the top quartile would likely represent a substantial portion of the traditional fast-fatigable motor units. It would be highly interesting to understand the recruitment and rate coding of the highest threshold motor units, at a minimum I would suggest using terms other than "entire range" or "full spectrum of recruitment thresholds"

Motor units were indeed identified between 0 and 80% of the maximal force in this study. This is due to the requirements of the decomposition algorithm that needs sustained and stable contraction to converge toward a set of separation vectors that generate sparse spike trains. Thus, it was not possible for our participants to sustain contractions above 80%MVC without generating fatigue.

However, it is important to note that only a few motor units are recruited above 80% of the maximal force in the TA (Van Cutsem et al., 1998, J Physiol), as well as in other muscles of the lower limb (Oya et al., 2009, J Physiol; Aeles et al., 2020, J Neurophysiol). Thus, we may have only missed a few motor units recruited above 80% of the maximal force. Nevertheless, we removed the terms ‘full spectrum of recruitment thresholds’ and ‘entire range’ from the manuscript to now read ‘most of the spectrum of recruitment thresholds observed in humans.’.

The quantification of hysteresis using torque appears to make self-evident the observation that lower threshold motor units demonstrate less hysteresis with respect to torque. If there is motor unit discharge there will be force. I believe this limitation goes beyond the floor effects discussed in the manuscript. Traditionally, individuals have used the discharge of a lower threshold unit as the measure on which to apply hysteresis analyses to infer ion channel function in human spinal motoneurons.

We agree with the reviewer that the hysteresis is classically estimated using the firing rate of a ‘reporter unit’ with the delta F method (introduced in humans by Gorassini et al..), or most recently with the advances in motor unit identification using the cumulative spike train of the identified motor unit. The researchers use this data as a proxy of the synaptic drive, and compare their values at recruitment and derecruitment thresholds of the ‘test unit’.

As mentioned above in response to reviewer 1, this approach was not possible in our study as we did not have the same units across contractions to estimate cumulative spike trains. It was therefore not possible to pool the data across contractions as we did here to generate force/firing rate relations on the widest range of force. This limitation is now highlighted in the discussion section (P19; L470): ‘This result must be confirmed with a more direct proxy of the net synaptic drive, such as the firing rate of a reference low-threshold motor neuron used in the delta F method (Gorassini et al., 1998), or the cumulative spike train of low-threshold motor neurons (Afsharipour et al., 2020).’.

The main findings are not entirely novel. See Monster and Chan 1977 and Kanosue et al 1979.

We agree with the reviewer that the results of the paper are remarkably aligned with previous experimental findings in humans, in animals, or with in vitro and in silico models. However, we believe that our study shows in humans the incredible variety of rate coding patterns within a pool of motor units that span most of the spectrum of recruitment thresholds observed in humans. It also highlights the variability of rate coding patterns between motor neurons that have a similar recruitment threshold. Finally, we observe differences between pools of motor neurons innervating two different muscles in the lower limb, mirroring what has been done in the past in the upper limb muscle.

**Recommendations for the authors:**

**Reviewer #1 (Recommendations For The Authors):**
The wording 'decode' across the manuscript may sound somewhat unsuitable for the context, because 'decode' would involve interpreting the signals and activities to understand how they relate to specific variables or proxies of behavior. Here in this study it does not necessarily involve the interpretation, but sounds to be used for decomposing the signal into the constituent motor units. As such, it might be appropriate to use other words such as decompose, read out, or extract.

‘Decode’ was removed from the manuscript to now read motor unit ‘identification’

**Reviewer #2 (Recommendations For The Authors):**
Figures 1 and 2 are informative and interesting. Figures 3 and 4 are harder to interpret. For example, in Figure 4, data plotted along the diagonal is overplotted and not as informative.

For the sake of clarity, we separated the lines of the fits and the scatter plots in in the right panels in Figure 3. In Figure 4, we remove the scatter plots and only reported the lines of the fits for each participant.

Do you think the different durations of the isometric plateau across contraction intensities influenced motor unit derecruitment? Longer duration in lower threshold motor units would have resulted in a larger effect of PICs?

We did not find an effect of the duration of the plateau on the derecruitment threshold. Notably, a computational study found that the duration of the plateau may impact the delta F, due to the combination of PICs, spike threshold accommodation and spike frequency adaptation (Revill & Fuglevand, 2011, J Neurophysiol). However, we did not use the delta F value here to estimate the effect of PICs on the hysteresis.

L703. For the measure of firing rate hysteresis the difference between recruitment and derecruitment was calculated, but why not use the delta-F method? This is more commonly used to assess hysteresis as a rough estimate of intrinsic dynamics.

As further discussed above, this approach was not possible in our study as we did not have the same units across contractions to estimate cumulative spike trains. It was therefore not possible to pool the data across contractions as we did here to generate force/firing rate relations on the widest range of force.

This was mentioned in the discussion (P19; L470):

‘This result must be confirmed with a more direct proxy of the net synaptic drive, such as the firing rate of a reference low-threshold motor neuron used in the delta F method (Gorassini et al., 1998), or the cumulative spike train of low-threshold motor neurons (Afsharipour et al., 2020).’

L144. The standard deviation seems high. Some participants had fewer than 20 motor units and your number of participants per muscle was eight, could you state the complete range?

A table was added in the results section to indicate the yields of the decomposition per contraction.

If other studies are able to randomly sample motor units with intramuscular electrodes does this also represent an estimate of rate coding from the 'entire' pool? One criticism of HDsEMG arrays is that they are biased towards decomposing superficial larger motor units and in the male sex.

The decomposition of EMG signals recorded with arrays of surface electrodes is indeed biased toward the identification of motor units with the larger action potentials in the signal (large and superficial; Farina & Holobar, 2016, Proceedings of IEEE). We took advantage of the latter limitation by performing successive contractions at different levels of force with the objective to identify the last recruited motor units (larger units according to the size principle), while tracking the smaller ones. In that way, we were able to sequentially identify motor units recruited from 0% to 75% of the maximal force. A similar approach could be applied to selective intramuscular electrodes. However, because identifying motor units up to maximal force requires a highly selective pair of fine wires or needle electrodes, the procedure described above should be repeated hundreds of times to reach the same samples as those obtained in our study.

L151-161. The ratio between simulated and decomposed surface EMG reached 55% for the TA and 70% for the VL. How does this provide support that the "entire" MU pool was sampled?

As said above, we do not identify all the motor units during each contraction, but rather the larger ones with the larger action potentials within the EMG signals. However, we used here a sequential approach to identify new motor units during each trial while tracking smaller units. In that way, we were able to sequentially identify on average 130 motor units per muscle.

To avoid any confusion, we removed the references to ‘*entire’* pools in the manuscript.

L266. How is it possible that in some participants no motor units were recruited below 5% of MVC? Do the authors suspect they produced force from synergist muscles or that the decomposition failed to identify these presumably smaller and deeper motor units?

This mostly results from the limitations of the decomposition algorithm. In these participants, it is likely that the decomposition was biased toward motor units only active during the plateau of force or recruited at the end of the ramp.

Figure 2B. Do the higher threshold motor units with linear responses receive more inhibitory input (coactivation) or are devoid of large PIC effects?Were antagonist muscles recorded? During higher contraction intensities, greater antagonist coactivation in some trials or participants may have linearized the firing rate profiles (e.g., Revill and Fuglevand, 2017).L427. This is a neat finding that higher threshold motor units are less likely to have the functional hallmark of a strong PIC effect and may therefore be more representative of extrinsic inputs. Could this be an advantage to increase the precision of stronger contractions or reduce the fatigability of muscle fibres during repeated strong contractions?

Synaptic contacts with Renshaw cells (Fyffe, 1991, J Neurophysiol) and Ia inhibitory interneurons (Heckman & Binder, 1991, J Neurophysiol) are widespread within pools of motor units, which induces homogeneously distributed inhibitory inputs. However, the amplitude of these inhibitory inputs can increase with muscle force. We found that the EMG amplitude of the soleus and the gastrocnemius medialis recorded with bipolar EMG during the dorsiflexion increased with the force. Therefore, the higher inhibitory at higher force may also contribute to the linearisation of the force/firing rate relations observed with high threshold motor neurons, as suggested by Revill and Fuglevand (2017, J Physiol).

We discussed this point in the new version of the manuscript (P17; L415):

‘The level of recurrent and reciprocal inhibition has also probably increased with the increase in force during the ramp up, progressively blunting the effect of persistent inward currents for late-recruited motor units (Kuo et al., 2003; Hyngstrom et al., 2007; Revill and Fuglevand, 2017). This may also explain the larger percentage of high-threshold motor units with a linear fit for the firing rate/force relation (Figure 2), as the integration of larger inhibitory inputs should linearise the firing rate/force relation (Revill and Fuglevand, 2017).’.

In Figure 2B, it makes sense that linear firing rate responses occur later in the ramp contraction when myotendinous slack is lower. Do the authors think contractile dynamics are matched to the firing rate profiles?

To our knowledge, there is no direct data on the link between the linearity of the force/firing rate relation and the stiffness of the tendon. A recent work from Mazzo et al. (2021, J Physiol) has shown that repeated stretches of calf muscles, which induce a decrease in their stiffness, induced an increase in motor unit firing rate at low levels of forces. This indicates that the contractile properties of the muscle may potentially also impact the profile of rate coding when considered as function of force.

We added this point in the discussion (P20; L512):

‘On a different note, the steep increase in firing rate over the first percentages of the ramp-up may also enable the motor units to produce the required level of force despite having a more compliant muscletendon unit (Mazzo et al., 2021).’

L371. It is likely that Marshall et al., 2022, recorded over 100 unique motor units from the same animal.

The reviewer is right that Marshall may have identified hundreds of motor units across sessions in one non-human primate. However, there is no ways to verify this statement as they used fine wire electrodes inserted in different locations in each session, which made it impossible to verify the uniqueness of each identified unit. Conversely, we verified in our study that all the motor units were unique using the distribution of their surface action potentials across the 236 surface electrodes.

L378. What do the authors mean by "rate coding is similar"? I find this statement confusing. Is this regarding the absolute firing rate range, response to force increases, hysteresis, or how they scale with contraction intensity?

This statement was removed from the discussion to avoid any confusion.

**Reviewer #3 (Recommendations For The Authors):**
The authors may want to consider other mechanisms of the linearization of discharge rates of medium and high threshold motor units. Monica's work may suggest that, over time, there is a subthreshold activation of the PIC, which serves to linearize the eventual suprathreshold activation underlying repetitive discharge. Additionally, Andy has shown that inhibitory drive from cutaneous inputs can linearize the initial acceleration of low threshold motor units - cutaneous inputs, or even Ib inputs, may be greater later in the contraction and serve to linearize discharge rates.

We thank the reviewer for their input on the discussion, where we now discuss this point:

‘The level of recurrent and reciprocal inhibition has also probably increased with the increase in force during the ramp up, progressively blunting the effect of persistent inward currents for late-recruited motor units (Kuo et al., 2003; Hyngstrom et al., 2007; Revill and Fuglevand, 2017). This may also explain the larger percentage of high-threshold motor units with a linear fit for the firing rate/force relation (Figure 2), as the integration of larger inhibitory inputs should linearise the firing rate/force relation (Revill and Fuglevand, 2017).’.

Lines 433 - intrinsic properties, in particular the afterhyperpolarization, will likely influence maximal discharge rate and provide a ceiling to the change in firing rate.

This point is now discussed in the draft (P17; L428):

‘This difference may be explained by smaller excitatory synaptic inputs onto low- than high-threshold motoneurons (Powers and Binder, 2001; Heckman and Enoka, 2012), lower synaptic driving potential of the dendritic membrane (Powers and Binder, 2000; Cushing et al., 2005; Fuglevand et al., 2015), and longer and larger afterhyperpolarisation phase in low- than high-threshold motoneurons (Bakels and Kernell, 1993; Gardiner, 1993; Deardorff et al., 2013; Caillet et al., 2022).’

The actual yield per contraction is not entirely clear. Figure S2 is quite nice in this regard, but a table with this and other information on it may be helpful. This would help with the beginning of the abstract and discussion when it is stated that, on average over 100 motor units were identified per person.

We added a table in the results to give the number of motor units identified per contraction.

Are the thin film units represented in S2 and S3?

Only motor units identified from signals recorded with arrays of surface electrodes are presented in figures S2 and S3.